# Neural retina-specific Aldh1a1 controls dorsal choroidal vascular development via Sox9 expression in retinal pigment epithelial cells

So Goto[1,2], Akishi Onishi[1,3]*, Kazuyo Misaki[4], Shigenobu Yonemura[4], Sunao Sugita[1,3], Hiromi Ito[1], Yoko Ohigashi[1], Masatsugu Ema[5], Hirokazu Sakaguchi[6], Kohji Nishida[2], Masayo Takahashi[1,3]

[1]Laboratory for Retinal Regeneration, RIKEN Center for Developmental Biology, Kobe, Japan; [2]Department of Ophthalmology, Osaka University Graduate School of Medicine, Suita, Japan; [3]Kobe City Eye Hospital Research Center, Kobe, Japan; [4]Ultrastructural Research Team, RIKEN Center for Life Science Technologies, Kobe, Japan; [5]Department of Stem Cells and Human Disease Models, Research Center for Animal Life Science, Shiga University of Medical Science, Otsu, Japan; [6]Department of Advanced Device Medicine, Osaka University Graduate School of Medicine, Suita, Japan

*For correspondence:
aonishi@cdb.riken.jp

Competing interests: The authors declare that no competing interests exist.

**Abstract** VEGF secreted from retinal pigment epithelial (RPE) cells is responsible for the choroidal vascular development; however, the molecular regulatory mechanism is unclear. We found that $Aldh1a1^{-/-}$ mice showed choroidal hypoplasia with insufficient vascularization in the dorsal region, although Aldh1a1, an enzyme that synthesizes retinoic acids (RAs), is expressed in the dorsal neural retina, not in the RPE/choroid complex. The level of VEGF in the RPE/choroid was significantly decreased in $Aldh1a1^{-/-}$ mice, and RA-dependent enhancement of VEGF was observed in primary RPE cells. An RA-deficient diet resulted in dorsal choroidal hypoplasia, and simple RA treatment of $Aldh1a1^{-/-}$ pregnant females suppressed choroid hypoplasia in their offspring. We also found downregulation of Sox9 in the dorsal neural retina and RPE of $Aldh1a1^{-/-}$ mice and RPE-specific disruption of Sox9 phenocopied $Aldh1a1^{-/-}$ choroidal development. These results suggest that RAs produced by Aldh1a1 in the neural retina directs dorsal choroidal vascular development via Sox9 upregulation in the dorsal RPE cells to enhance RPE-derived VEGF secretion.
DOI: https://doi.org/10.7554/eLife.32358.001

## Introduction

The retina is the light-sensitive tissue at the back of the eye that consists of photoreceptor cells, retinal pigment epithelium (RPE), and its basement (Bruch's) membrane. Disorder of the retina causes vision loss. The primary nutrient source for the retina is the choroid (*Bill et al., 1980*), which is a highly vascularized tissue layer surrounding the retina. The choroid consists of three layers: Haller's layer (large blood vessel layer), Sattler's layer (medium size blood vessels), and the choriocapillaris (*Hogan et al., 1971*; *Nickla and Wallman, 2010*). The choriocapillaris is a unique anastomosed vascular structure with an extraretinal fenestrated capillary bed that lies in a single plane below Bruch's membrane.

In the mouse retina, choroidal development begins from embryonic day (E)13.5 (*Marneros et al., 2005*; *Saint-Geniez et al., 2006*). Vascular endothelial growth factor (VEGF) is known to be a central regulator of vascular development during embryogenesis (*Coultas et al., 2005*), and VEGF secreted

**eLife digest** The retina is the part at the back of our eyes that detects light and sends this information to our brain. Within the retina is a layered structure containing the light-sensitive cells, known as the neural retina, and another protective layer of cells called the retinal pigment epithelium. A surrounding network of blood vessels, the choroid, keeps the retina healthy by supplying oxygen and nutrients. When the choroid does not work properly, eye disease can result. A common example is age-related macular degeneration, where blood vessels in the choroid either break down or start growing uncontrollably in the wrong places. In both cases, light-sensitive cells are damaged and eventually die. This causes vision loss that worsens over time.

The choroid forms early in life, within the developing embryo. The retinal pigment epithelium helps the choroid to develop properly by producing a protein, VEGF, which supports the growth of blood vessels. However, it was not clear what signals tell this tissue to start making VEGF in the first place and then to keep making it.

To address this, Goto et al. looked at eye development in mutant mice that lack an enzyme called Aldh1a1. This enzyme's role is to make a molecule called retinoic acid, which is known to be vital for many biological processes including the growth and development of embryos. Aldh1a1 is not made in the choroid of normal mice, just in the neural retina. Yet the choroid in the mutant mice without Aldh1a1 still grew fewer blood vessels than normal. Their retinal pigment epithelium also produced less VEGF and had lower levels of a protein called Sox9, which is known to make the gene for VEGF more active.

Goto et al. went on to show that simply removing retinoic acid from the diet of normal mice produced the same choroid defect as in the mutant mice with no Aldh1a1. Genetically manipulating otherwise normal mice to decrease the levels of Sox9 in the retinal pigment epithelium had a similar effect. In contrast, giving Aldh1a1-deficient mice extra retinoic acid or artificially increasing their levels Sox9 was enough to make the choroid develop normally. These experiments showed that retinoic acid produced in the neural retina directs choroid development by making Sox9 more active, which in turn encourages the retinal pigment epithelium to produce VEGF.

These findings bring new insights into the molecular signals that control choroid development. In the future, they may also help scientists to better understand why blood vessels in the choroid become abnormal in eye diseases like age-related macular degeneration.

DOI: https://doi.org/10.7554/eLife.32358.002

from RPE cells is indispensable for the vascular development and maintenance of the choroid (*Saint-Geniez et al., 2009*; *Le et al., 2010*; *Kurihara et al., 2012*). VEGF expression is enhanced by a number of transcription factors such as hypoxia-induced factor-1$\alpha$ (HIF-1$\alpha$), estrogen-related receptor-$\alpha$ (ERR$\alpha$), and peroxisome proliferator-activated receptor gamma corecceptor 1$\alpha$ (PGC-1$\alpha$) (*Ziello et al., 2007*; *Ueta et al., 2012*). However, the molecular mechanism of choroidal vascular development and the regulatory mechanism of VEGF secreted from RPE have not been clarified.

In the present study, we found that aldehyde dehydrogenase one family, member A1 (*Aldh1a1*) knockout (*Aldh1a1$^{-/-}$*) mice have hypoplasia in the dorsal region of the choroid. Aldh1a1 is an enzyme that synthesizes retinoic acids (RAs) and is expressed in the dorsal neural retina from the embryonic stage, not in the RPE and the choroid (*Matt et al., 2005*; *Molotkov et al., 2006*; *Luo et al., 2006*; *Kumar et al., 2012*). RAs are essential for biological activities such as reproduction, development, growth, and maintenance (*Kam et al., 2012*; *Rhinn and Dollé, 2012*; *Cunningham and Duester, 2015*). We analyzed choroidal vascular development in *Aldh1a1$^{-/-}$* mice, and demonstrated how Aldh1a1 expressed in the neural retinas in trans enhances VEGF secretion from the RPE to the choroid. We also found that, mechanistically, Sox9 expression in RPE is downstream of the signaling pathway mediated by Aldh1a1 in the neural retina.

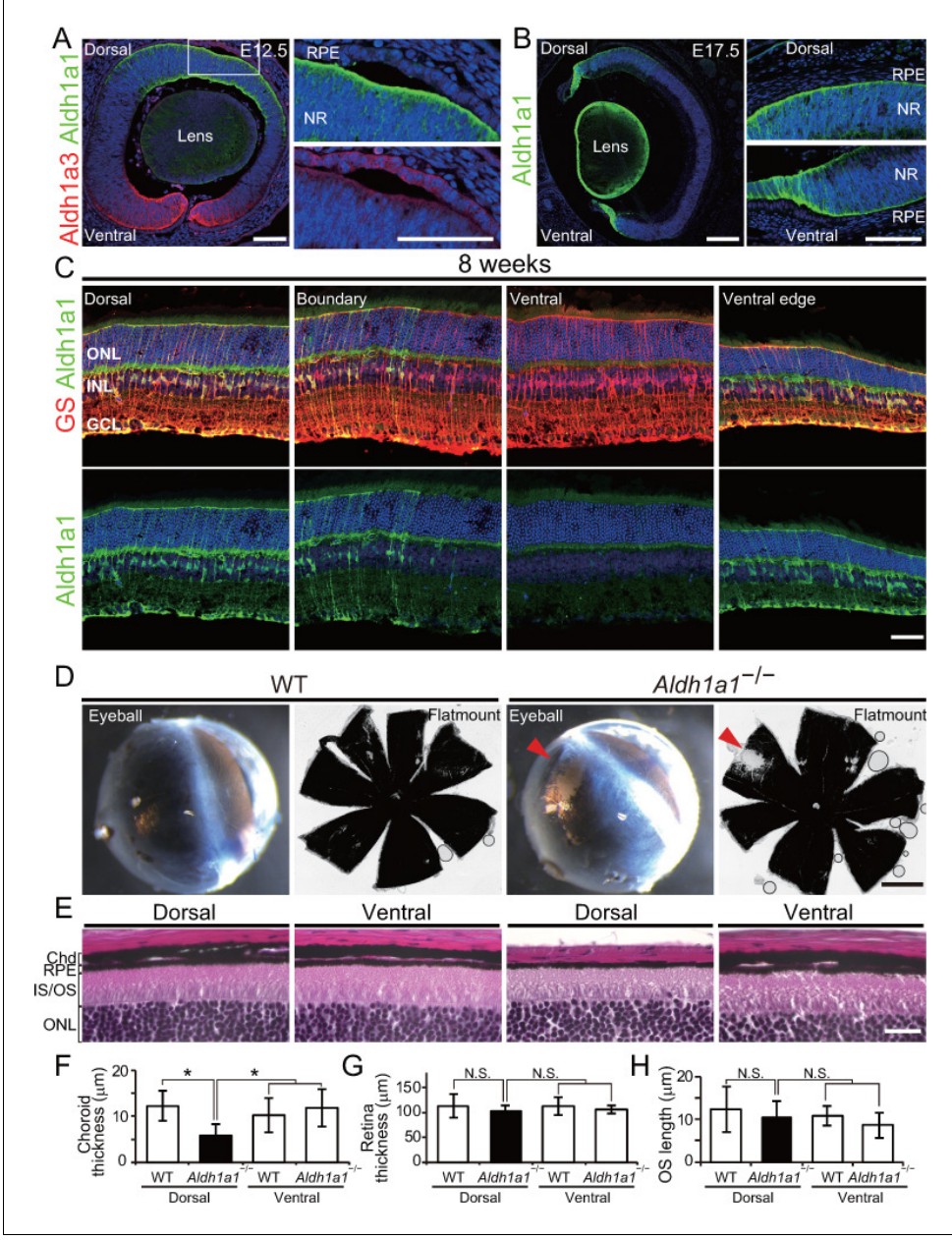

**Figure 1.** Aldh1a1 is predominantly expressed in the dorsal neural retina during embryonic and adult stages, and *Aldh1a1*[−/−] mice have less pigmentation and a thin choroid in the dorsal area. (A–C) Section immunohistochemistry of the mouse retinas labeled with Aldh1a1, Aldh1a3 and glutamine synthetase (GS) antibodies. (A) At E12.5, Aldh1a1 (green) and Aldh1a3 (red) are expressed in dorsal and ventral neural retina (NR), respectively. At the dorsal edge (white box), Aldh1a1 is expressed only in NR, not in RPE (right upper panel), while some RPE cells are Aldh1a3-positive (right lower panel). (B) At E17.5, Aldh1a1 (green) is expressed in the dorsal half and the ventral edge of the retina, but no Aldh1a1-positive RPE cells were detected in the dorsal (right upper panel) and ventral (right lower panel) regions. (C) At 8 weeks, Aldh1a1 (green)-positive cells were double-labeled with GS (red) at the dorsal, boundary, and ventral edge regions (see *Figure 1—figure supplement 1A, B*; for the 'boundary' region). ONL, outer nuclear layer; INL, inner nuclear layer; GCL, ganglion cell layer. (D) Dissected eyeballs and choroidal flat-mounts of adult (8-week-old) wild-type (WT) and *Aldh1a1*[−/−] mice. Loss of pigmentation was observed in the dorsonasal region (red arrowhead) of *Aldh1a1*[−/−] eyes. (E) Hematoxylin and eosin (H and E) staining of WT and *Aldh1a1*[−/−] retinal sections. The pigmented layer (choroid) of the dorsal *Aldh1a1*[−/−] retina is thinner than those on the other sides. ONL, outer nuclear layer; IS/OS, inner segment/outer segment; RPE, retinal pigment epithelium; Chd, choroid. (F–H) Quantitative evaluation of choroidal thickness (F), retinal thickness (G), and the length of outer

*Figure 1 continued on next page*

*Figure 1 continued*

segments (**H**) of the H and E-stained retinal sections. Data represent the average ±SD; n = 8–9 from 4 to 5 mice per group. *p<0.05. N.S., not significant. [Scale bars, 50 μm (**A and C**), 200 μm (**B**), 1 mm (**D**), and 20 μm (**E**).].
DOI: https://doi.org/10.7554/eLife.32358.003

The following source data and figure supplement are available for figure 1:

**Source data 1.** Source Data for *Figure 1F–H*.
DOI: https://doi.org/10.7554/eLife.32358.005

**Figure supplement 1.** In situ hybridization (ISH) with the *Aldh1a1* probe in the adult (8-week-old) mouse retina.
DOI: https://doi.org/10.7554/eLife.32358.004

## Results

### Aldh1a1 is preferentially expressed in the dorsal region of the mouse neural retinas, not in the RPE/choroid complex

It was previously reported that Aldh1a1 is expressed in the dorsal region of embryonic and adult retinas (*McCaffery et al., 1991*; *Fan et al., 2003*; *Matt et al., 2005*). However, the retinal cell types that express Aldh1a1 have not been precisely identified. Therefore, we performed immunohistochemistry and in situ hybridization studies in embryonic and adult mouse retinas.

At E12.5, Aldh1a1 was localized in the progenitor cells at the dorsal neural retina, and no Aldh1a1 expression was observed in the RPE and choroid complex (*Figure 1A*). Aldh1a3, an Aldh1 family enzyme that generates RA, was mainly localized in the ventral region of the retina and RPE. At E17.5, Aldh1a1 expression was observed in the dorsal region and the ventral edge, but no expression was detected in the RPE/choroid complex (*Figure 1B*). In adult tissue, in situ hybridization showed that Aldh1a1-positive cells were preferentially localized in the middle of the inner nuclear layer (INL) cells, and had spread to the dorsal region of the retina (*Figure 1—figure supplement 1A and B*). We also observed Aldh1a1-positive cells at the ventral edge of the adult retina (*Figure 1C*). Immunohistochemistry of sectioned tissues showed Aldh1a1 colocalized with glutamine synthetase (GS), a marker of Müller glia. These results indicate that Aldh1a1 is preferentially expressed in the retinal progenitor cells of the dorsal region from the embryonic stage, and in Müller cells in the dorsal region and the ventral edge of the retina in adulthood.

Next, we analyzed the morphological differences between wild-type (WT) and $Aldh1a1^{-/-}$ eyes. In adult $Aldh1a1^{-/-}$ mice, the eyeballs displayed a loss of pigmentation in the dorsonasal area (*Figure 1D*). Hematoxylin and eosin (H and E) staining of the paraffin sections revealed that pigmentation was lost from the choroidal region, not from RPE cells (*Figure 1E*). The choroidal thickness was significantly less than that of the dorsal and ventral sides of WT eyes and the ventral side of $Aldh1a1^{-/-}$ eyes (*Figure 1F*). There were no obvious morphological differences in the thickness of the neural retina or in the length of the outer segments of photoreceptor cells (*Figure 1G and H*), as reported previously (*Fan et al., 2003*). These findings suggest that the choroidal vessels of $Aldh1a1^{-/-}$ mice are hypoplastic.

### Aldh1a1$^{-/-}$ mice exhibited choroidal hypoplasia in the dorsal region

To characterize choroidal vascularization in the $Aldh1a1^{-/-}$ mice, we first immunostained choroidal flat-mounts for endomucin antibody, a specific marker for choriocapillaris, and isolectin B4, which mainly visualizes choroidal medium-sized/large blood vessels. Surprisingly, the $Aldh1a1^{-/-}$ mice exhibited choroidal hypoplasia in the dorsal region of the eyes (*Figure 2A*), although Aldh1a1 is normally expressed in the neural retina. In the dorsal and ventral regions of WT and the ventral region of $Aldh1a1^{-/-}$ eyes, the choriocapillaris was normal, with a dense mesh structure, whereas fewer choroidal vessels with more avascular areas were detected in the dorsal region of $Aldh1a1^{-/-}$ eyes (*Figure 2B and C*). We also performed whole-mount immunohistochemistry for ZO-1, a tight junction marker of RPE cells (*Figure 2D*). In the dorsal region of $Aldh1a1^{-/-}$ eyes, the vascular density was significantly lower than that in the other regions (*Figure 2E*), whereas there were no significant morphological differences in the localization of ZO-1 signals and RPE size between WT and $Aldh1a1^{-/-}$ eyes (*Figure 2F*). Taken together, these results demonstrate that $Aldh1a1^{-/-}$ eyes show choroidal hypoplasia without degeneration of the RPE in the dorsal region.

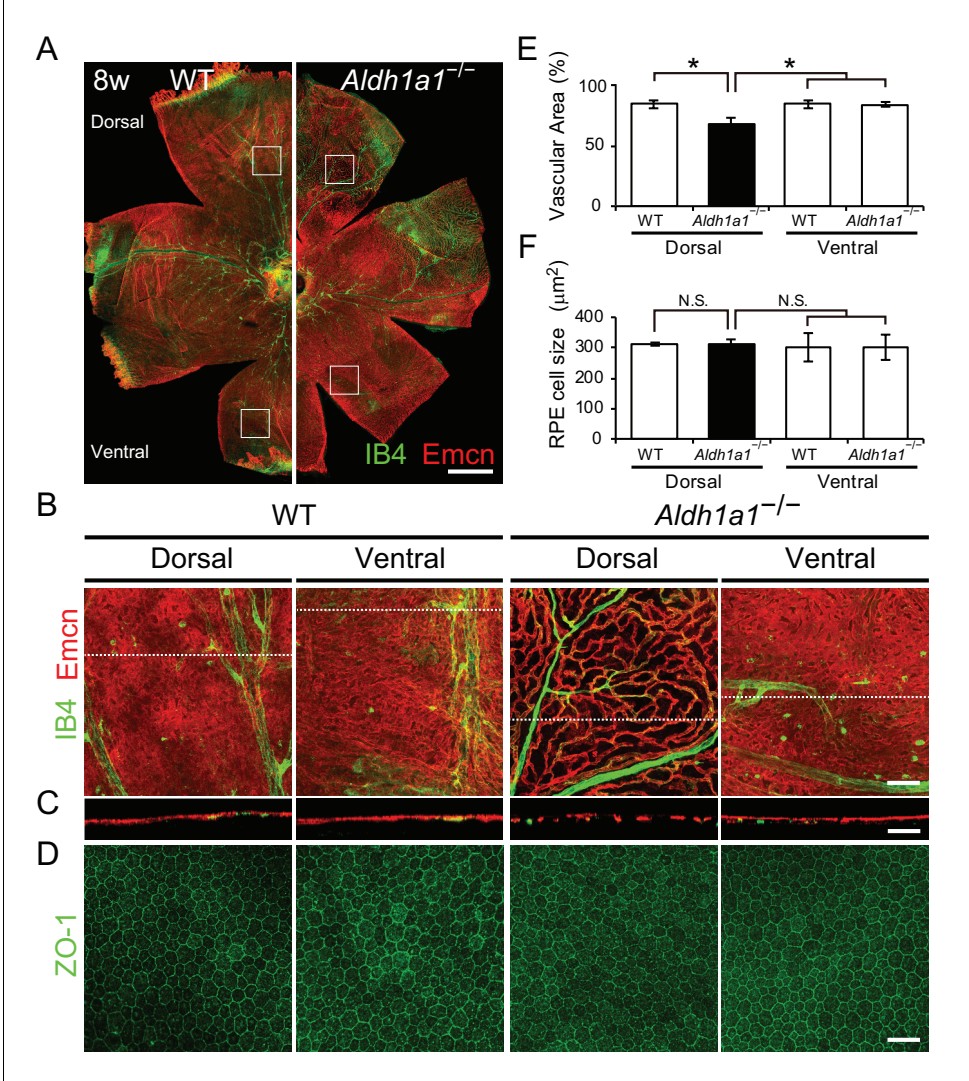

**Figure 2.** Hypoplasia at the dorsal side of choroid in *Aldh1a1*⁻/⁻ mice. (**A**) Representative choroidal flat-mount immunohistochemistry of adult (8-week-old) WT (left panel) and *Aldh1a1*⁻/⁻ (right panel) posterior eyes stained for endomucin (Emcn, red) and isolectin B4 (IB4, green). (**B**) High-magnification Z-stack images of the choroidal flat-mounts collected from the dorsal and ventral areas of WT and *Aldh1a1*⁻/⁻ mice (four white boxes shown in (**A**)) stained with Emcn (red) and IB4 (green). The dorsal image of *Aldh1a1*⁻/⁻ mice indicates poor vascularization. (**C**) Orthogonal images of the Z-stacks (broken lines in (**B**)) showing breaks/holes in the dorsal region of *Aldh1a1*⁻/⁻ eyes stained with Emcn antibody (red) and IB4 (green). (**D**) RPE flat-mount immunohistochemistry of WT and *Aldh1a1*⁻/⁻ stained with ZO-1 (green). (**E and F**) Quantitative evaluation of the vascular density and size of RPE cells of adult WT and *Aldh1a1*⁻/⁻ mice. Data represent the average ± SD; n = 8 per group. *p<0.05. N.S., not significant. [Scale bars, 500 μm (**A**), 50 μm (**B–D**).].

DOI: https://doi.org/10.7554/eLife.32358.006

The following source data is available for figure 2:

**Source data 1.** Source Data for *Figure 2E,F*.
DOI: https://doi.org/10.7554/eLife.32358.007

To investigate further the morphological features of the choroidal vasculature and the neural retina/RPE/Bruch's membrane complex, sections of WT and *Aldh1a1*⁻/⁻ eyes were examined by transmission electron microscopy (TEM). The TEM images demonstrated that *Aldh1a1*⁻/⁻ eyes exhibited apparently normal morphology of photoreceptor and RPE cells, despite the complete absence of choroidal pigmentation and the presence of a thin dorsal choroid layer (*Figure 3A*). Bruch's

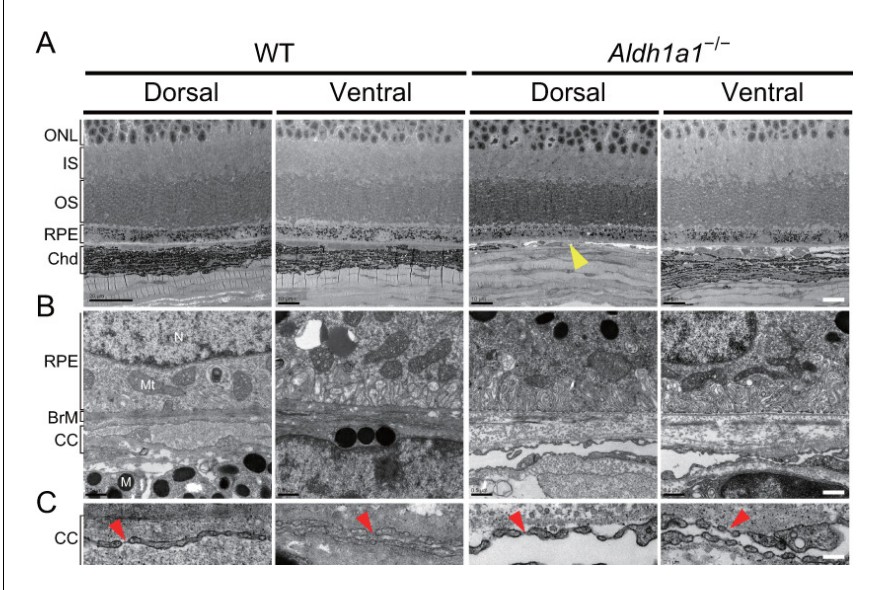

**Figure 3.** Choroidal vessels of *Aldh1a1*$^{-/-}$ mice have features of capillaries. (**A**) Electron micrographs of the dorsal and ventral areas from adult (8-week-old) WT and *Aldh1a1*$^{-/-}$ mice. The absence of choroidal pigmentation and a thin choroid layer is observed in the dorsal section of *Aldh1a1*$^{-/-}$ mice (yellow arrowhead). (**B**) Higher magnification of electron micrographs of retinal pigment epithelium (RPE) and Bruch's membrane (BrM). There are no differences between WT and *Aldh1a1*$^{-/-}$ mice in the morphology of RPE and BrM. M, melanocyte; Mt, mitochondria; N, nucleus. (**C**) The higher magnification of electron micrography allows visualization of the fenestrations in the choriocapillaris. Representative fenestrated structures are indicated by red arrowheads. In the dorsal section of *Aldh1a1*$^{-/-}$ mice, the choriocapillaris shows fenestrations similar to those in the dorsal and ventral sides of WT and the ventral side of *Aldh1a1*$^{-/-}$ mice. [Scale bars, 10 μm (**A**), 0.5 μm (**B**), and 0.1 μm (**C**).].
DOI: https://doi.org/10.7554/eLife.32358.008

The following figure supplement is available for figure 3:

**Figure supplement 1.** Choroidal vessels of the *Aldh1a1*$^{-/-}$ mice have both Flt1 and Kdr expression.
DOI: https://doi.org/10.7554/eLife.32358.009

membrane did not differ significantly between WT and *Aldh1a1*$^{-/-}$ eyes (*Figure 3B*). Remarkably, the choroidal vessels in the hypoplastic dorsal section of *Aldh1a1*$^{-/-}$ eyes showed fenestrations, a characteristic of capillaries, as did those in the dorsal and ventral regions of WT and the ventral region of *Aldh1a1*$^{-/-}$ eyes (*Figure 3C*). Taken together, these results suggest that the hypoplastic blood vessels of the *Aldh1a1*$^{-/-}$ eyes maintain the characteristics of the choriocapillaris, including an intact RPE and Bruch's membrane.

## Aldh1a1$^{-/-}$ choroidal vessels exhibited an abnormal distribution of VEGF receptor subtypes

Although several reports have suggested that FMS-like tyrosine kinase 1 (Flt1, also called VEGF receptor 1) and kinase insert domain protein receptor (Kdr, also called VEGF receptor 2) are expressed in choroidal vessels (*Zhao and Overbeek, 2001*; *Witmer et al., 2003*; *Cross et al., 2003*; *Saint-Geniez et al., 2006*), the blood vessel type-specific receptor expression remains ambiguous. Therefore, we examined the regional distribution of *Flt1*-DsRed and *Kdr*-EGFP in choroidal vessels in WT and *Aldh1a1*$^{-/-}$ eyes by using *Flt1*-BAC-DsRed;*Kdr*-BAC-EGFP mice (*Matsumoto et al., 2012*). In the choroidal flat-mounts from adult WT mice, the choriocapillaris coexpressed both *Flt1*-DsRed and *Kdr*-EGFP, while the medium-sized/large choroidal vessels expressed only Flt1-DsRed (*Figure 3—figure supplement 1A–D*). In addition, more *Flt1*-DsRed appeared to be expressed in the medium-sized/large choroidal vessels than in the choriocapillaris (*Figure 3—figure supplement 1D*). In contrast, in the dorsal region of *Aldh1a1*$^{-/-}$ eyes, all the blood vessels associated with the hypoplastic choroid coexpressed both *Flt1*-DsRed and *Kdr*-EGFP (*Figure 3—figure*

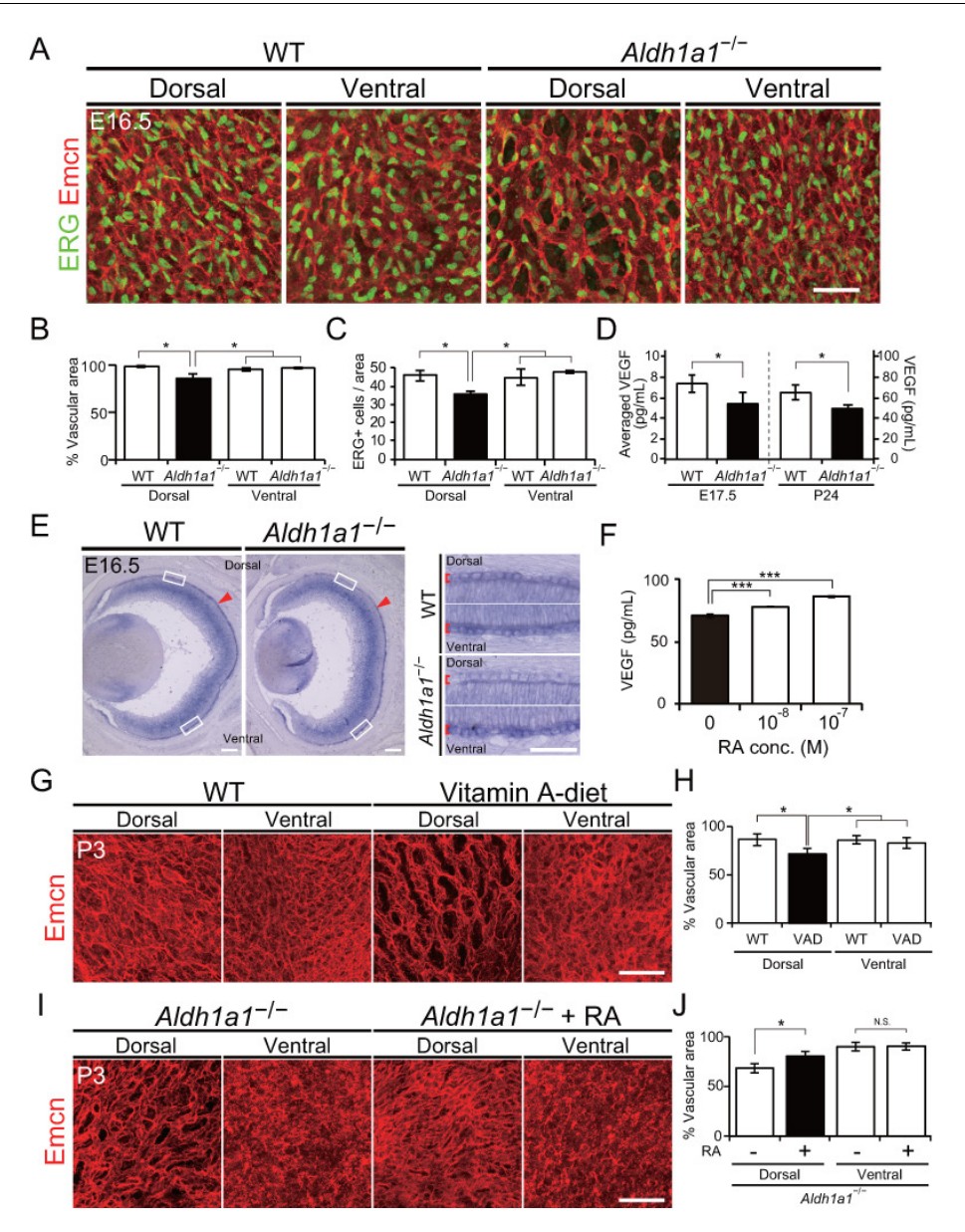

**Figure 4.** Retinoic acids modulate VEGF secretion by RPE cells. (**A**) Choroidal flat-mount immunohistochemistry of E16.5 WT and *Aldh1a1⁻/⁻* embryos stained for endomucin (Emcn, red) and ETS-related gene (ERG, green). Note that hypovascularization and fewer vascular endothelial cells were observed in the dorsal region of *Aldh1a1⁻/⁻* embryonic eyes. (**B and C**) Quantitative evaluation of the density of Emcn-positive vessels and the number of ERG-positive cells in E16.5 WT and *Aldh1a1⁻/⁻* embryos. Data represent the average ±SD; n = 4 per group. *p<0.05. (**D**) ELISA analysis of VEGF secreted from WT and *Aldh1a1⁻/⁻* RPE-choroid complex at E17.5 and P24. Data represent the average ±SD; n = 4 independent samples per group. *p<0.05. (**E**) In situ hybridization on E16.5 WT and *Aldh1a1⁻/⁻* eyes with the *Vegfa* probe (DIG-labeled, purple). *Vegfa* expression was reduced in the dorsal RPE cells of *Aldh1a1⁻/⁻* eyes (upper RPEs from red arrowhead). Right panels show the higher magnification images of left panels (four white boxes) and red square brackets indicate RPE layer. (**F**) Retinoic acid (RA)-dependent enhancement of VEGF secretion of human primary RPE cells evaluated by ELISA. Data are means three times ELISA determinations. ***p<0.001. (**G**) Choroidal flat-mount immunohistochemistry of P3 WT and Vitamin-A-deficient (VAD) mice stained for endomucin (Emcn, red). Dorsal choroidal hypoplasia was observed in VAD mice. (**H**) Quantitative evaluation of the vascular density of P3 WT and VAD mice. Data represent the average ±SD; n = 5–6 per group. *p<0.05. (**I**) Choroidal flat-mount of P3 *Aldh1a1⁻/⁻* and *Aldh1a1⁻/⁻* mice from a mother treated with RA by oral gavage between E10 and E16, immunostained with anti-endomucin antibody (Emcn, red) This RA

*Figure 4 continued on next page*

*Figure 4 continued*

treatment of the mother restored the choroidal vascularization in P3 *Aldh1a1*$^{-/-}$ pups. (**J**) Quantitative evaluation of the vascular density in *Aldh1a1*$^{-/-}$ mice and *Aldh1a1*$^{-/-}$ mice from a mother treated with RA. Data represent the average ± SD; n = 4 per group. *p<0.05. N.S., not significant. [Scale bars, 50 μm (A, right panels in E, (**G and I**), 200 μm (left panels in E)].

DOI: https://doi.org/10.7554/eLife.32358.010

The following source data and figure supplements are available for figure 4:

**Source data 1.** Source Data for *Figure 4B–D,F,H,J*.
DOI: https://doi.org/10.7554/eLife.32358.013
**Figure supplement 1.** Developmental flat-mount immunohistochemical analysis of choroid in the *Aldh1a1*$^{-/-}$ mice.
DOI: https://doi.org/10.7554/eLife.32358.011
**Figure supplement 2.** Neural retina-specific conditional disruption of *Aldh1a3* did not show choroidal hypoplasia in the ventral retina.
DOI: https://doi.org/10.7554/eLife.32358.012

---

*supplement 1B–D*). Also, the level of expression of *Flt1*-tdsRed in the *Aldh1a1*$^{-/-}$ blood vessels appeared lower than that in the medium-sized/large choroidal vessels.

## Aldh1a1-mediated RA production regulates dorsal choroidal vascular development by enhancing VEGF secretion from RPE cells

We next explored the developmental time point at which choroidal hypoplasia appears in *Aldh1a1*$^{-/-}$ mice. Since the formation of the choroidal vascular network starts at approximately E13.5 (*Zhao and Overbeek, 2001*; *Marneros et al., 2005*; *Saint-Geniez et al., 2006*), we compared the vascular development in the choroid of the eyes of WT and *Aldh1a1*$^{-/-}$ mice from E16.5 to postnatal (P)28 using a choroidal flat-mounts immunostained with antibodies against endomucin and ETS-related gene (ERG), which are enriched in endothelial cells (*Figure 4A*; *Figure 4—figure supplement 1*). In the dorsal region of *Aldh1a1*$^{-/-}$ eyes, endomucin immunostaining was patchy, indicating that choroidal hypoplasia was already detectable (*Figure 4A and B*). There were significantly fewer endothelial cells in the area than in WT eyes (*Figure 4C*).

Based on studies showing that VEGF secreted from RPE cells is necessary for choroidal vascular development (*Sakamoto et al., 1995*; *Saint-Geniez et al., 2009*; *Le et al., 2010*), we hypothesized that RAs synthesized by Aldh1a1 in the neural retina stimulate the RPE to enhance VEGF secretion to the basal side of the choroid. To test this hypothesis, we first evaluated the VEGF level in E17.5 and P24 RPE-choroid complexes by ELISA. VEGF levels in the *Aldh1a1*$^{-/-}$ RPE-choroid complex were significantly lower than those in WT eyes at both time points (*Figure 4D*). In addition, in situ hybridization revealed that *Vegfa* mRNA expression of the E16.5 *Aldh1a1*$^{-/-}$ RPE cells was reduced at the dorsal half where Aldh1a1 is expressed in the WT neural retina (*Figure 4E*, shown as arrowheads). Next, we tested RA-dependent VEGF secretion by the primary human RPE cells. We measured VEGF in the culture medium after RA treatment. As a result, RAs significantly enhanced VEGF secretion in a dose-dependent manner (*Figure 4F*).

Because RA is the active metabolite of vitamin A (*Shams et al., 1993*; *Amengual et al., 2012*), we generated vitamin A-deficient (VAD) mice by feeding a vitamin A-deficient diet (*Chihara et al., 2013*). At P3, VAD mice showed dorsal choroidal hypoplasia in the flat-mount analysis (*Figure 4G*). In the dorsal region of VAD eyes, the vascular density was significantly lower than that in the other regions such as the dorsal and ventral regions of WT and the ventral region of VAD eyes (*Figure 4H*). Also, RA administration to pregnant *Aldh1a1*$^{-/-}$ mice by oral gavage from E10 to E16 significantly suppressed the dorsal choroidal hypoplasia in their offspring (*Figure 4I and J*). These results indicate that RA controls dorsal choroidal vascular development and that dorsal choroidal hypoplasia in *Aldh1a1*$^{-/-}$ mice is causally related to a deficiency in RA synthesis.

These observations raised the question of whether Aldh1a3, another RA-producing enzyme in the mouse retina, also affects choroidal vascularization, because Aldh1a3 begins to be expressed in the entire RPE cell layer and in the ventral region of the neural retina from E10.5 (*Matt et al., 2005*). To test this possibility, we conditionally disrupted *Aldh1a3* in the neural retina (floxed *Aldh1a3* mice crossed with *Pax6*-α-Cre), however, in these mice, choriocapillaris development was found to be normal (*Figure 4—figure supplement 2*).

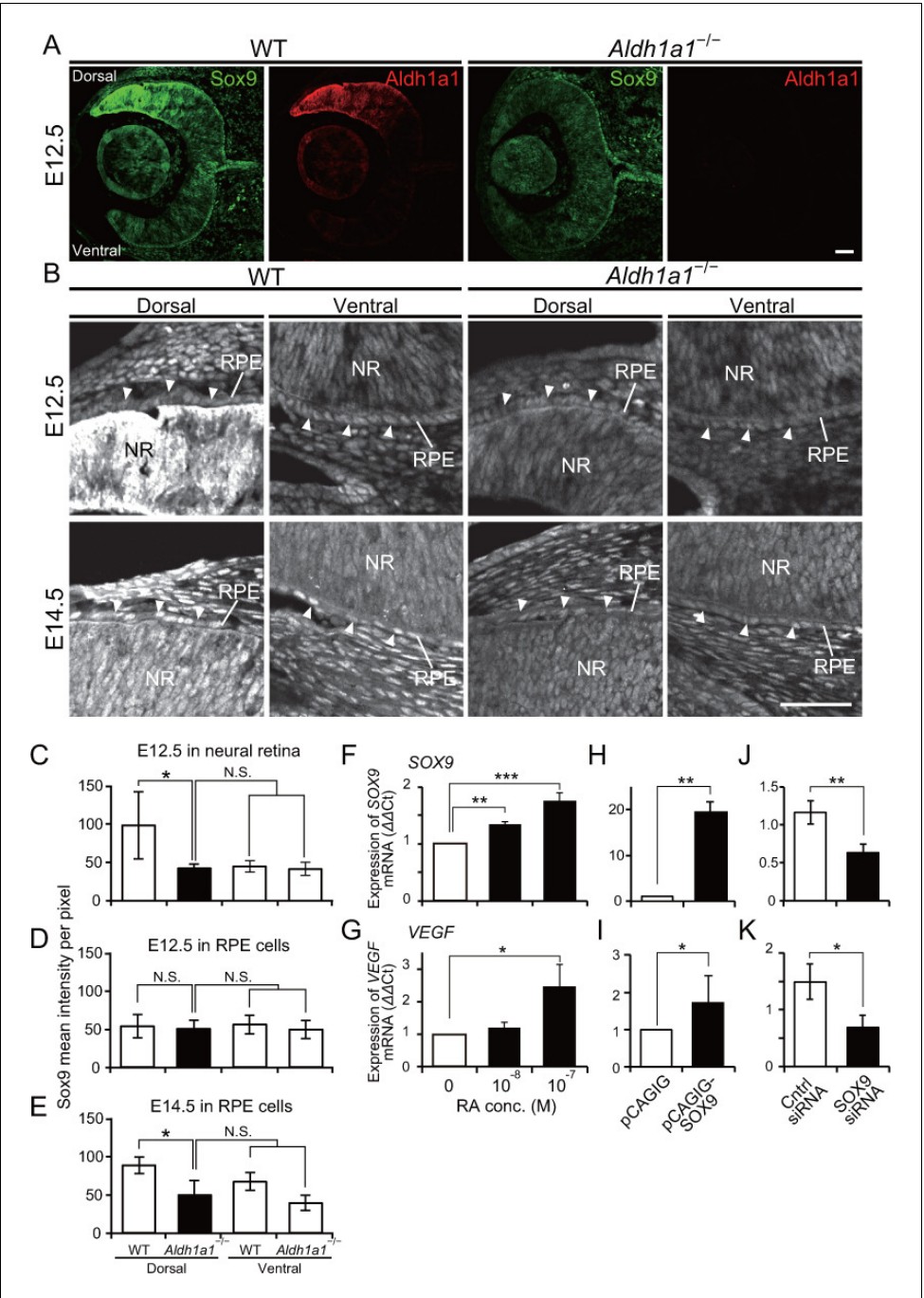

**Figure 5.** Sox9 expression is downregulated in RPE cells of *Aldh1a1*−/− mice. (**A**) Sox9 (green) and Aldh1a1 (red) staining of WT and *Aldh1a1*−/− eyes at E12.5. (**B**) Sox9 (white) expression in neural retina and RPE (white arrowheads) of WT and *Aldh1a1*−/− eyes at E12.5 (upper panels) and E14.5 (lower panels). Sox9 was strongly expressed in the dorsal neural retinas of E12.5 WT eyes, and downregulated in RPE cells of E14.5 *Aldh1a1*−/− eyes. [Scale bars, 50 µm (**A** and **B**).]. (**C–E**) Quantitative evaluation of the Sox9 immunofluorescence intensity in embryonic WT and *Aldh1a1*−/− eyes. Sox9 intensity was quantified in the E12.5 neural retina (**C**), E12.5 RPE cells (**D**), and E14.5 RPE cells (**E**). Data represent the average ±SD; n = 4 per group. *p<0.05. N.S., not significant. (**F–K**) *Sox9* and *Vegfa* mRNA expression in primary RPE cells in response to RA exposure (**F and G**), Sox9 overexpression (**H and I**), and Sox9 knockdown (**J and K**). Relative expression of *Sox9* mRNA (**F, H, and J**) and *Vegfa* mRNA (**G, I, and K**) normalized to β-*actin* mRNA are shown. Data are representative of three experiments. *p<0.05, **p<0.01, ***p<0.001. N.S., not significant.

DOI: https://doi.org/10.7554/eLife.32358.014

*Figure 5 continued on next page*

*Figure 5 continued*

The following source data and figure supplements are available for figure 5:

**Source data 1.** Source Data for *Figure 5C–K*.

DOI: https://doi.org/10.7554/eLife.32358.016

**Figure supplement 1.** Pax6 expression in the developing RPE cells of WT and *Aldh1a1*$^{-/-}$ mice.

DOI: https://doi.org/10.7554/eLife.32358.015

**Figure supplement 1—source data 1.** Source Data for *Figure 5—figure supplement 1B and C*.

DOI: https://doi.org/10.7554/eLife.32358.017

In conclusion, RAs derived from Aldh1a1 but not Aldh1a3 control choroidal vascular development by enhancing VEGF secretion from RPE cells.

## Sox9 is downregulated in both the dorsal RPE and neural retina in the Aldh1a1$^{-/-}$ mice

We next investigated the transcription factors that directly enhance VEGF expression in dorsal RPE cells. Recently, it was reported that conditional disruption of Pax6 in the RPE at the early embryonic stages resulted in the loss of pigmentation (*Raviv et al., 2014*), and Pax6 conditional knockout mice in which the *Vegfa* promoter is synergistically transactivated by Pax6 and Sox9 exhibit choroidal hypoplasia (*Cohen et al., 2016*). Therefore, we performed immunohistochemistry to detect Pax6 and Sox9 in sections of embryonic WT and *Aldh1a1*$^{-/-}$ retinas. The intensity of Pax6 immunofluorescence in the dorsal *Aldh1a1*$^{-/-}$ RPE was slightly lower than WT at E12.5 and E14.5, but did not show a significant difference (*Figure 5—figure supplement 1A–C*). Next, we measured the developmental expression of Sox9 (*Figure 5A and B*). In the E12.5 WT neural retina, Sox9 was predominantly expressed in the dorsal region rather than in the ventral region, and the expression level at the dorsal region became comparable to the ventral region at E14.5. In *Aldh1a1*$^{-/-}$ neural retinas, Sox9 immunofluorescence in the dorsal region was reduced as much as that of the ventral region (*Figure 5A and C*). In the E12.5 WT RPE cells, there was no difference in Sox9 immunofluorescence between dorsal and ventral region, and the intensity increased at E14.5. In the E12.5 *Aldh1a1*$^{-/-}$ RPE cells, the immunofluorescence was comparable to WT, but was significantly lower than that of E14.5 WT (*Figure 5B,D and E*). These densitometry results suggest that Aldh1a1 enhances Sox9 expression in the dorsal neural retina and RPE cells during development.

To determine whether Sox9 enhances VEGF in RPE cells in an RA-dependent manner, we measured *Sox9* and *Vegfa* mRNA expression in primary RPE cells in response to RA exposure. The results showed that both *Sox9* and *Vegfa* mRNAs (*Figure 5F and G*) were enhanced in an RA-dependent manner. To examine whether Sox9 regulates *Vegfa* in RPE cells, we performed overexpression and knockdown experiments. Overexpression of *Sox9* by transient transfection of a pCAGIG-Sox9 vector resulted in upregulation of *Vegfa* mRNA (*Figure 5H and I*). In contrast, knockdown by transient transfection of *Sox9* siRNA resulted in downregulation of *Vegfa* mRNA (*Figure 5J and K*). Taken together, these results strongly suggest that Sox9 enhanced by Aldh1a1-mediated RA upregulates *Vegfa* expression in RPE cells.

## Conditional disruption of Sox9 in RPE cells phenocopies choroidal hypoplasia in the Aldh1a1$^{-/-}$ mice

We next explored further whether the Aldh1a1-driven Sox9 expression in the dorsal neural retina and RPE is involved in choroidal vascular development. To generate mice with selective deletion of *Sox9* in the developing RPE or neural retina, mice with a conditional deletion of *Sox9* (*Sox9*$^{flox/flox}$; *Kist et al., 2002*) were mated with either *Tyr*-Cre (RPE-cKO of *Sox9*; *Delmas et al., 2003*) or *Pax6*-α-Cre (Retina-cKO of *Sox9*; *Marquardt et al., 2001*), respectively. In the Cre-reporter assay with R26R-H2B-mCherry mice (*Abe et al., 2011*) on an albino background, mCherry expression from E16.5 *Tyr*-Cre mice was observed in all RPE, although a few mCherry-positive cells were found in the neural retina (*Figure 6—figure supplement 1A*). Also, mCherry expression in *Pax6*-α-Cre mice was restricted to the dorsal and ventral portions of the neural retina as reported previously (*Marquardt et al., 2001*), but no mCherry-positive cells were found in the choroid (*Figure 6—figure supplement 1B*). In RPE-cKO of *Sox9*, we found less pigmentation in the dorsal region, and

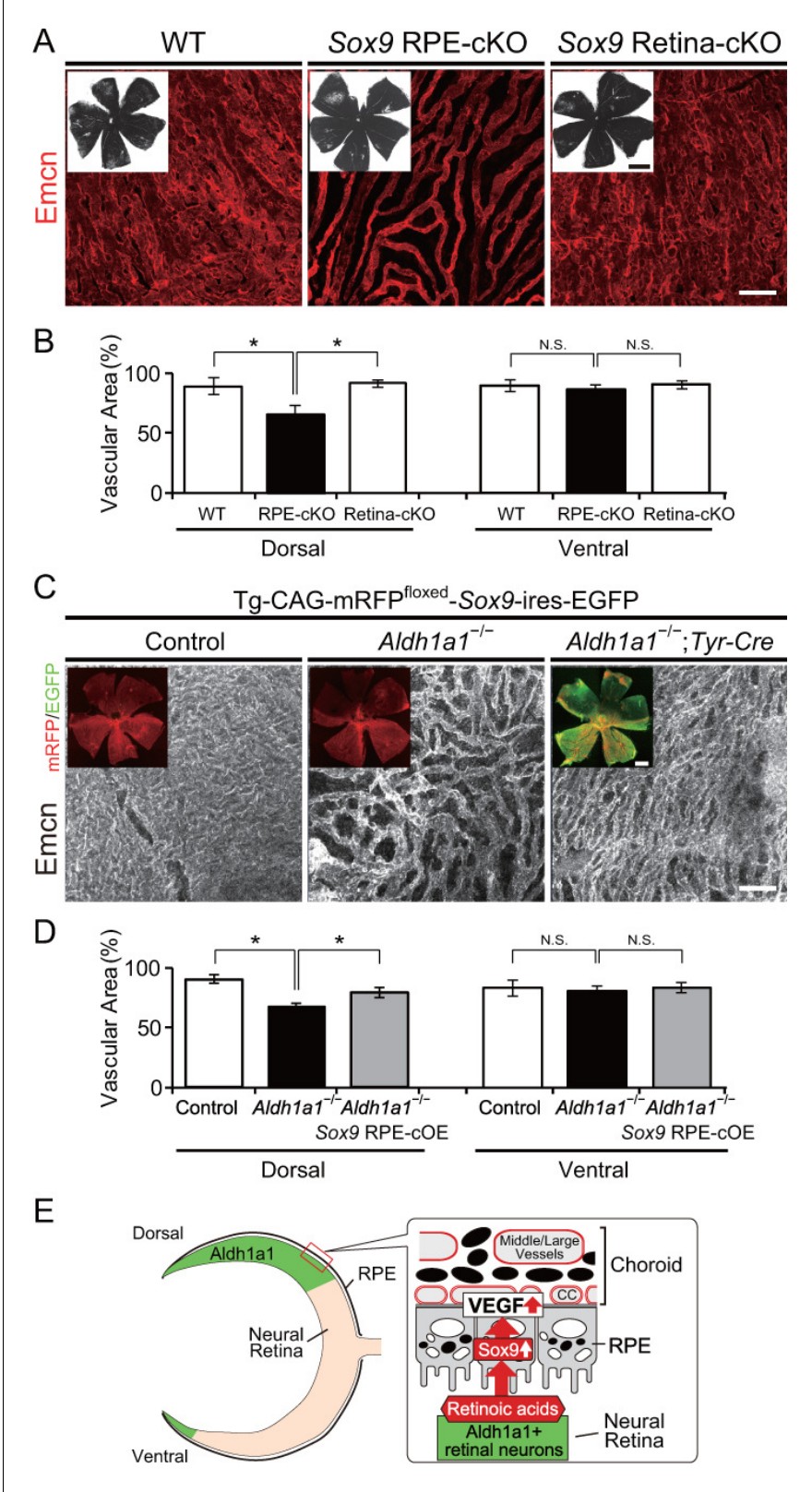

**Figure 6.** RPE-derived Sox9 controls choroidal vasculature development. (**A**) Endomucin expression (Emcn, red) in 5-week-old choroidal flat-mounts. RPE-specific conditional knockout of *Sox9* (*Sox9* RPE-cKO) mimicked the choroidal hypoplasia seen in *Aldh1a1*⁻/⁻ mice (middle panel), but retina-specific conditional knockout of *Sox9* (*Sox9* Retina-cKO) did not (right panel). Insets represent each choroidal flat-mount. (**B**) Quantitative evaluation of

*Figure 6 continued on next page*

*Figure 6 continued*

the vascular density of 5-week-old WT, *Sox9* RPE-cKO, and *Sox9* Retina-cKO. Data represent the average ±SD; n = 4–5 per group. *p<0.05. N.S., not significant. (**C**) Endomucin (Emcn, white) expression in P7 choroidal flatmounts. The choroid was hypovascularized in Tg-CAG-mRFP$^{floxed}$-SOX9-ires-EGFP;*Aldh1a1*$^{-/-}$ eyes (middle panel). Choroidal hypoplasia in *Aldh1a1*$^{-/-}$ mice was rescued in *Aldh1a1*$^{-/-}$;RPE-specific conditional overexpression of *Sox9* (*Aldh1a1*$^{-/-}$;*Tyr*-Cre, right panel). Insets represent reporter (mRFP and EGFP) expression. (**D**) Quantitative evaluation of the vascular density of P7 control, *Aldh1a1*$^{-/-}$, and *Aldh1a1*$^{-/-}$;*Sox9* RPE-cOE. Data represent the average ±SD; n = 5 per group. *p<0.05. N.S., not significant. [Scale bars, 1 mm (insets in A), 50 μm (**A and C**), 500 μm (insets in C).]. (**E**) Model summarizing that neural retina-specific Aldh1a1 controls choroidal vascularization in the dorsal region. Retinoic acids (RA) synthesized by Aldh1a1 regulate Sox9 expression, and then Sox9 enhances VEGF secretion from the dorsal RPE cells.

DOI: https://doi.org/10.7554/eLife.32358.018

The following source data and figure supplement are available for figure 6:

**Source data 1.** Source Data for *Figure 6B and D*.

DOI: https://doi.org/10.7554/eLife.32358.020

**Figure supplement 1.** Cre reporter assay of Tyr-Cre and Pax6-α-Cre mice using albinized R26R-H2B-mCherry mice.

DOI: https://doi.org/10.7554/eLife.32358.019

---

significantly poorer vasculature in the dorsal choroidal area than the other areas (*Figure 6A and B*), which phenocopied *Aldh1a1*$^{-/-}$ eyes. Interestingly, although *Sox9* was disrupted in all RPE cells, the poor vasculature phenotype was restricted to the dorsal region, and did not appear in the ventral region (*Figure 6B*). Conversely, Retina-cKO of *Sox9* showed no hypoplasia of the choroidal vasculature or lack of pigmentation (*Figure 6A and B*), indicating that Sox9 in the neural retina is not responsible for choroidal vascular development.

Next, we attempted to rescue the choroidal hypoplasia phenotypes imposed by *Aldh1a1* deletion by restoring Sox9 signaling using Cre-inducible *Sox9*-overexpressing mice (RPE-cOE of *Sox9*, *Kim et al., 2011*). *Tyr*-Cre-induced overexpression of Sox9 in these mice significantly recovered the choroidal hypoplasia phenotypes in the dorsal region (*Figure 6C and D*). These results indicated that Sox9 expression enhanced by Aldh1a1 in developing RPE cells is critical for normal choroidal vasculature development.

## Discussion

The molecular mechanism underlying the development and function of the neural retina has been well studied, but that responsible for the development of the choroidal vessels has not been thoroughly investigated, despite the pathophysiological importance of these structures in human eye disease. Here, we could immunohistochemically discriminate choriocapillaris from medium-sized/large blood vessels in Haller's/Sattler's layer in mouse eyes and then demonstrated that disruption of *Aldh1a1*, which is predominantly localized in the dorsal neural retina, resulted in choroidal hypoplasia in the dorsal portion of the eyes.

Previous developmental and electrophysiological analyses of the *Aldh1a1*$^{-/-}$ mouse has shown that the neural retina is apparently normal (*Fan et al., 2003*). By broadening the phenotyping scope, we identified vascular hypoplasia in the dorsal choroid of *Aldh1a1*$^{-/-}$ eyes. Aldh1a1 is expressed in dorsal retinal progenitor cells from E10.5 (*Matt et al., 2005*), and we observed no Aldh1a1 expression in either RPE cells or choroid in E12.5 and E17.5 mouse eyes (*Figure 1A and B*). CD31-positive endothelial cells and VEGF expression in mouse eyes can be observed in choroid and primitive RPE cells at E10.5 (*Saint-Geniez et al., 2006*), and choroidal vascular development is initiated at E11.5 when periocular vessels emerge following formation of the vascular network at around E13.5 (*Zhao and Overbeek, 2001*; *Marneros et al., 2005*; *Saint-Geniez et al., 2006*). The requirement for RPE-derived VEGF during embryonic development was reported using RPE-specific *Vegfa*-knockout mice (*Vegf*$^{flox/flox}$;*VMD2*-Cre mice), in which Cre expression can be conditionally induced in RPE cells by doxycycline administration (*Le et al., 2008*). Although conditional disruption of *Vegfa* in RPE cells at E10 or E13 resulted in decreased choroidal vascular density, loss of the RPE-produced VEGF after E15 caused no significant defects in the choroidal vasculature (*Le et al., 2010*). Together with our

**Table 1.** Oligonucleotides used in this study

| Mouse primers | | Oligonucleotide sequences (5′—3′) |
| --- | --- | --- |
| Aldh1a1⁻/⁻ | Forward | TGAGCAAATCCTCCACAGCCCTGTTC |
| | Reverse | CTGCTAAAGCGCATGCTCCAGACTG |
| Aldh1a3 flox | Forward | TCTCTGACCAGCTTTCCAACCTTCAG |
| | Reverse | CTCAAACCAGCACCACCTCCATATTG |
| Sox9 flox | Forward | TCAGCAAGACTCTGGGCAAGCTCT |
| | Reverse | CTCAAAATCTGAGCCACTCCCTC |
| Tyr-Cre, Pax6-α-Cre | Forward | CCTGGAAAATGCTTCTGTCCGT |
| | Reverse | GTGTCCACATAGTCATTGGCAGAGTG |
| Sox9-Tg, Kdr-BAC-EGFP | Forward | AGCTGACCCTGAAGTTCATCTG |
| | Reverse | GTCGTCCTTGAAGAAGATGGTG |
| Flt1-BAC-tdsRed | Forward | GCTGCAGGCGCGGAGAAGGGCTCTC |
| | Reverse | CTTCACGTACACCTTGGAGC |
| IL2 (internal control) | Forward | GCCTAGAAGATGAACTTGGACCTCTG |
| | Reverse | GTGGAAGGATTCACTTGCACAGTGAC |
| Human RPE primers | | |
| Sox9 | Forward | CGTACCCGCACTTGCACAAC |
| | Reverse | TCTCGCTCTCGTTCAGAAGTC |
| Vegfa | Forward | TGCCCGCTGCTGTCTAAT |
| | Reverse | TCTCCGCTCTGAGCAAGG |
| β-actin | Forward | CCAACCGCGAGAAGATGA |
| | Reverse | CCAGAGGCGTACAGGGATAG |

DOI: https://doi.org/10.7554/eLife.32358.021

demonstration that choroidal VEGF production is downregulated in Aldh1a1⁻/⁻ embryos, these data suggest that Aldh1a1 expression during E10 to E13 in trans potentiates VEGF secretion from RPE cells to allow the development of normal choroidal vascularization. In addition, it should be noted that choroidal VEGF level was still reduced at P24 in the Aldh1a1⁻/⁻ mice. Considering that severe choriocapillaris vasoconstriction occurs in adult tamoxifen-induced Vegf^flox/flox;VMD2-Cre mice (**Kurihara et al., 2012**), it is possible that Aldh1a1 is also responsible for maintenance of the choriocapillaris.

We also demonstrated that RAs enhanced VEGF secretion from primary RPE cells and that VAD mice showed choroidal hypoplasia in the dorsal region. Also, RA administration suppressed the choroidal hypoplasia phenotype in Aldh1a1⁻/⁻ mice. These results strongly suggest that Aldh1a1-mediated RA is responsible for normal choroidal vascular formation by controlling VEGF secretion from RPE cells. Moreover, we observed downregulation of Sox9 in the dorsal Aldh1a1⁻/⁻ eyes. RPE-specific disruption of Sox9 replicated the phenotype of Aldh1a1⁻/⁻ eyes, and Sox9 overexpression in the Aldh1a1⁻/⁻ RPE cells rescued dorsal choroidal hypoplasia. Considering that in primary RPE cells RA exposure enhances both Sox9 and Vegfa expression and that overexpression and knockdown of Sox9 influences Vegfa expression, it is more likely that Aldh1a1-mediated RA production stimulates Sox9 expression in dorsal RPE cells and Sox9 then transactivates the Vegfa promoter. Retinoic acid receptor α (RARα) and retinoid X receptor α (RXRα) are expressed in RPE cells at an early embryonic stage (**Mori et al., 2001**). It is plausible that RARα and RXRα enhance Sox9 expression in dorsal RPE cells, although the precise molecular mechanism remains to be investigated.

We observed interactions between Aldh1a1, Sox9, and Vegfa genes; however, the results raised the question of why VAD mice and RPE-cKO of Sox9 show choroidal hypoplasia only in the dorsal region because the reduction of vitamin A and Sox9 also occurs in the ventral region. We do not yet have the answer, but one possibility is that ventral choroidal vascular development is governed by a different molecular pathway. For example, Aldh1a3, another RA-producing enzyme in the mouse

retina, is expressed in RPE cells from E10.5 to E12.5 and in the ventral neural retina during embryogenesis (*Matt et al., 2005*), *Figure 1B*). However, conditional disruption of *Aldh1a3* in the neural retina (floxed *Aldh1a3* mice crossed with *Pax6-α*-Cre) did not result in either choroidal hypoplasia or loss of pigmentation in the ventral region (*Figure 4—figure supplement 2*), suggesting that Aldh1a3-mediated RAs from the ventral neural retina are unlikely to affect ventral choroidal development. In addition, we did not observe upregulation of Sox9 in the ventral RPE cells from E12.5 to E14.5 (*Figure 5D and E*), suggesting that Sox9 in the ventral RPE cells is unnecessary for ventral choroidal development. Given that mRFP and GFP fluorescent intensity seems different between dorsal and ventral RPEs in Tg-CAG-mRFP$^{floxed}$-*SOX9*-ires-EGFP mice (*Figure 6C*), the difference might be the result of different developmental or physiological characteristics.

In summary, we demonstrate a novel role of Aldh1a1 in dorsal choroidal vascular development. The results of the present study strongly suggest that RA production resulting from Aldh1a1 expression in the dorsal neural retina upregulates Sox9 expression in the dorsal RPE cells to enhance RPE-derived VEGF secretion (*Figure 6E*). In addition, our results suggest that embryonic RA exposure may regulate future dorsal choroidal vessels in the adult. Because vascular hypoplasia could result in hypoxia and impaired nutrient supply, which could affect adjacent RPE and photoreceptor cells, future studies should investigate the degeneration of RPE and photoreceptor cells. For example, age-related macular degeneration (AMD) is the leading cause of severe vision loss in humans (*de Jong, 2006*) and is caused by abnormalities of the subfoveal choriocapillaris. *Aldh1a1*$^{-/-}$ and VAD mice would be useful not only to study regional differences in choroidal development and maintenance but also to explore both a risk factor and a potential therapeutic target for AMD, because the RA concentration in the neural retina is affected by various environmental factors such as dietary intake of Vitamin A and light irradiation.

# Materials and methods

## Animals

All animal experiments were conducted with the approval of the RIKEN Center for Developmental Biology Ethics Committee (No. AH18-05-23). Timed pregnant CD1 and C57BL/6 mice were purchased from the Laboratory for Animal Resources and Genetic Engineering, RIKEN Center for Developmental Biology. *Aldh1a1*$^{-/-}$ mice (*Fan et al., 2003*) were purchased from Jackson Laboratory and mated with CD-1 mice to allow visualization of the choroidal vessels. *Flt1*-BAC-DsRed;*Kdr*-BAC-EGFP mice (*Matsumoto et al., 2012*) were mated with CD-1 or albinized *Aldh1a1*$^{-/-}$ mice. To conditionally disrupt *Aldh1a3* in neural retinas, *Aldh1a3*-flox mice (*Dupé et al., 2003*) were mated with *Pax6-α*-Cre mice (*Marquardt et al., 2001*). To conditionally disrupt Sox9 in RPE cells and neural retinas, *Sox9*-flox mice (*Kist et al., 2002*) were mated with *Tyr*-Cre (*Delmas et al., 2003*) and *Pax6-α*-Cre mice, respectively. To conditionally overexpress Sox9 in RPE cells, CAG-mRFP1$^{floxed}$-*Sox9*-ires-EGFP transgenic mice (*Kim et al., 2011*) were mated with *Tyr*-Cre mice. Albinized R26R-H2B-mCherry mice (*Abe et al., 2011*) were used for the Cre reporter assay of *Tyr*-Cre and *Pax6-α*-Cre mice (*Figure 6—figure supplement 1A and B*). PCR primers used for genotyping are listed in *Table 1*. VAD diet feeding and RA treatments were performed as described previously (*Chihara et al., 2013*; *Fan et al., 2003*). Eight-week-old CD-1 mice were fed a vitamin A-deficient (VAD) diet (AIN93G-D13110GC; Research Diets, New Brunswick, USA). After 16 weeks of feeding this diet, the animals were used as breeding pairs. Pregnant mice received the VAD diet until 3 days postpartum. *All-trans*-RA (Sigma) was suspended in ethanol (5 mg/ml) and then either diluted in sunflower oil (125 μg/ml) and administered by oral gavage to pregnant females (2 mg/kg of body weight) every 12 hr from E10 to E16.

## Immunohistochemistry

For cryosections, embryos were fixed in 4% paraformaldehyde (PFA), cryoprotected in 30% sucrose in phosphate-buffered saline (PBS) overnight at 4°C, embedded in OCT compound (Tissue Tec; Sakura Fine Technical, Japan), and sectioned at 10 μm using a cryostat (HM560; Thermo Fisher Scientific, Waltham, MA). Specimens were blocked with horse serum for 1 hr at room temperature and incubated with primary antibodies overnight at 4°C, followed by incubation with secondary antibodies for 1 hr at room temperature.

For flat-mount immunostaining of the RPE/choroid, the tissues were fixed with 4% PFA at room temperature for 30 min, and washed three times with PBS containing 0.5% Triton X-100 (PBST, Nacalai Tesque, Japan), incubated with primary antibodies overnight at 4°C, followed by incubation with secondary antibodies for 1 hr at room temperature (*Zhu et al., 2012*). The primary antibodies and dilutions used were as follows: goat anti-Aldh1a1 (1:1,000; Abcam), rabbit anti-Aldh1a3 (1:1,000; Sigma), rat anti-endomucin (1:400; Millipore), mouse anti-GS (1:1,000; Millipore), FITC-isolectin B4 (1:100; Vector Laboratories), rabbit anti-ZO-1 (1:100; Invitrogen), rabbit anti-GFP (1:1,000; Abcam), rabbit anti-ERG (1:400; Abcam), rabbit anti-Pax6 (1:200; BioLegend), and rabbit anti-Sox9 (1:200; Millipore).

## Electron microscopic analysis

Mice were euthanized and perfused with ice-cold 4% PFA. Eyes were fixed with 2% glutaraldehyde in 4% PFA overnight. After washing with PBS, the eyes were postfixed with ice-cold 1% $OsO_4$ in 0.1 M sodium cacodylate buffer, pH 7.3, for 2 hr. The samples were then rinsed with distilled water, stained with 0.5% aqueous uranyl acetate for 2 hr or overnight at room temperature, dehydrated with ethanol and propylene oxide, and embedded in Poly/Bed 812 (Polyscience). Ultrathin sections were cut, double-stained with uranyl acetate and Reynolds' lead citrate, and viewed with a JEM 1010 or JEM 1400 transmission electron microscope (JEOL) at an accelerating voltage of 100 kV.

## Analysis of images

Labeled cells were imaged using an LSM 780 confocal microscope (Carl Zeiss). Images were processed using Photoshop CS2 software (Adobe Systems). Choroidal vascular density was analyzed using ImageJ software (NIH) as described previously (*Le et al., 2010*). Fluorescence intensity was quantified using Zen Black software (version 2012; Carl Zeiss) according to the manufacturer's instructions. All images shown are representative of three to eight independent experiments.

## In situ hybridization

In situ hybridizations were performed as described previously (*Acloque et al., 2008*).

## Enzyme-linked immunosorbent assay for VEGFA

The RPE/choroid was separated from the mouse eye and was homogenized and resolved in 100 µl RIPA buffer containing a protease inhibitor. The total VEGFA protein (pg) per mg of the extracted RPE/choroid tissue was calculated using ELISA development kits (R and D Systems, Minneapolis, MN) (*Ueta et al., 2012*).

## Cell culture, RA treatment, and transfection

Primary human RPE cells (Lonza) were maintained in Dulbecco's minimal essential medium (DMEM) supplemented with 5% fetal bovine serum (FBS) without antibiotics. Before RA treatment or transfection experiments, the RPE cells were suspended at $2 \times 10^5$ cells per well of a 6-well plate and cultured in DMEM supplemented with 5% FBS without antibiotics. After overnight culture, the medium was changed to RA-supplemented DMEM without FBS, and the cells were harvested after 24 hr incubation. For overexpression, 1 µg of pCAGIG and pCAGIG-Sox9 expression vectors (Addgene #11159) were transfected using Lipofectamine 3000 (Invitrogen) according to the manufacturer's instructions. For knockdown, control and human Sox9 siRNAs (Santa Cruz) were transfected using siRNA Transfection Reagent (Santa Cruz) according to the manufacturer's instructions. After 7 hr incubation, the medium was changed to DMEM without FBS (for overexpression) or DMEM plus 10% FBS (knockdown), and cells were harvested after another 24 hr incubation to quantify *Sox9* and *Vegfa* mRNA by reverse transcription–quantitative polymerase chain reaction (RT-qPCR).

## Reverse transcriptase–quantitative polymerase chain reaction

RT–qPCR was performed as described in a previous report (*Sugita et al., 2015*). Briefly, *Sox9*, *Vegfa*, and β-*actin* expression was analyzed in triplicate samples using a LightCycler model 480 (Roche Diagnostics), qPCR MasterMix (Roche Diagnostics), and highly specific Universal ProbeLibrary assays (Roche Diagnostics). The tested primers are described in *Table 1*, and the Universal Probes used were Probe#61 (*Sox9* and *Vegfa*), and Probe#64 (β-*actin*). Relative mRNA expression was normalized

to ΔΔCt of β-*actin* using relative quantification software (Roche Diagnostics). Results were reported as the relative expression of each molecule (ΔΔCt: control cells = 1).

## Statistical analysis

All data are presented as the means ± SD. JMP Pro version 10.0.2 (SAS Institute Inc.) was used for statistical analysis, and data were analyzed using analysis of variance (ANOVA) followed by the Tukey–Kramer multiple-comparison test. When only two groups were compared, a two-sided Student's *t*-test was used.

## Acknowledgements

We received generous support from all members of the Takahashi Laboratory. We thank Laboratory for Animal Resources and Genetic Engineering, RIKEN Center for Developmental Biology for providing and maintaining mice, and RIKEN BioResource Center, Drs. R Morita (RIKEN CDB), H Fujiwara (RIKEN CDB), Y Kubota (Keio University), R Ashery-Padan (Tel Aviv University), H Akiyama (Gifu University), P Chambon (IGBMC), and S Mader (Université de Montréal) for providing mice. This work was supported in part by grants from Japan Agency for Medical Research and Development (grant number 17bm0204002h0005), JSPS KAKENHI (grant number 24687010 and 17K11471), and grants-in-aid of The Kato Memorial trust for NAMBYO Research. SG was financially supported from RIKEN by a Junior Research Associate (JRA) program for graduate students.

## Additional information

### Funding

| Funder | Grant reference number | Author |
|---|---|---|
| Japan Agency for Medical Research and Development | 17bm0204002h0005 | Masayo Takahashi |
| Japan Society for the Promotion of Science | 24687010 | Akishi Onishi |
| Japan Society for the Promotion of Science | 17K11471 | Akishi Onishi |
| The Kato Memorial trust for NAMBYO Research | | Akishi Onishi |

The funders had no role in study design, data collection and interpretation, or the decision to submit the work for publication.

### Author contributions

So Goto, Conceptualization, Resources, Data curation, Formal analysis, Funding acquisition, Validation, Investigation, Visualization, Methodology, Writing—original draft, Project administration; Akishi Onishi, Conceptualization, Resources, Data curation, Formal analysis, Supervision, Funding acquisition, Validation, Investigation, Visualization, Methodology, Writing—original draft, Project administration, Writing—review and editing; Kazuyo Misaki, Data curation, Formal analysis, Investigation, Visualization, Methodology; Shigenobu Yonemura, Data curation, Supervision, Investigation, Visualization, Methodology; Sunao Sugita, Data curation, Formal analysis, Investigation, Methodology; Hiromi Ito, Yoko Ohigashi, Resources, Data curation, Formal analysis, Validation, Investigation; Masatsugu Ema, Resources; Hirokazu Sakaguchi, Supervision; Kohji Nishida, Supervision, Project administration, Writing—review and editing; Masayo Takahashi, Conceptualization, Supervision, Funding acquisition, Project administration, Writing—review and editing

### Author ORCIDs

So Goto http://orcid.org/0000-0003-4171-0781
Akishi Onishi http://orcid.org/0000-0002-2775-6567

## Ethics

Animal experimentation: All animal experiments were conducted with the approval of the RIKEN Center for Developmental Biology Ethics Committee (No. AH18-05-23)

## Decision letter and Author response

Decision letter https://doi.org/10.7554/eLife.32358.029
Author response https://doi.org/10.7554/eLife.32358.030

## Additional files

**Supplementary files**
• Transparent reporting form
DOI: https://doi.org/10.7554/eLife.32358.022

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
