## [Decision Letter]

Thank you for submitting your article "Neural retina-specific *Aldh1a1* controls dorsal choroidal vascular development via *Sox9* expression in RPE cells" for consideration by *eLife*. Your article has been favorably evaluated by Marianne Bronner as the Senior Editor, Jeremy Nathans as the Reviewing Editor and three reviewers. The following individual involved in review of your submission has agreed to reveal her identity: Jing Chen (Reviewer #3).

As you will see, all of the reviewers were impressed with the importance of your work, but they also had a number of specific comments.

I am including the three reviews at the end of this letter, as there are a variety of specific and useful suggestions in them. Most of the comments are focused on greater documentation of the ocular phenotypes. One point that may be more challenging to address in a reasonable time frame is the specificity of the *Tyr*-Cre and *Pax6*-α-Cre lines. This can be done by crossing these Cre lines to a reporter (a nuclear localized reporter might be best). This point is summarized as item 4 in the comments of reviewer #2, which I am reproducing below.

"One concern relates to the foundations for the mechanistic conclusions of the paper. The model and main interpretations leading to the model are based on genetic manipulations using Cre-lines that the authors claim to be specific, i.e., *Tyr*-Cre, specific for RPE, and *Pax6*-α-Cre, specific for neural retina. However, since the specificities of these are critical for the conclusions of the paper, the authors need to better characterize them. Certainly, these lines have been published before but they were not described in the retina/choroid context. For instance, how specific is the *Tyr*-Cre line to the RPE? Since the choriocapillaris has melanin, it is possible that the manipulations of *Sox9*, either the KO or the overexpression rescue experiment, are not totally RPE-specific. This might affect the authors' interpretations in terms of the mechanism. Note that *Sox9* is also expressed in the choroid (Figure 4). Is *Pax6*-α-Cre expressed in all neural retina?"

The challenge is made greater because the best design of this experiment would be to cross the Cre lines to reporters in an albino background (to avoid the difficulty of visualizing fluorescent reporters or immunofluorescent staining in heavily pigmented cells, i.e. RPE and choroid).

*Reviewer #1:*

In this study, the authors link neuroretinal *Aldh1a1* to dorsal choroidal vascular development via regulation of RPE synthesis of VEGF. The authors found a compelling phenotype of hypoplasia choroidal with insufficient vascularization in the dorsal region in *Aldh1a1* knockout mice. The authors found that *Aldh1a1*, an enzyme used for retinoic acid synthesis, is localized in the dorsal neuroretina from the very early embryonic stage to adult stage. By using in situ techniques and conditional knockout mice, the authors found that *Aldh1a1*-induced RA in neural retinal directs dorsal choroidal vascularization via RPE-derived VEGF production by increasing *Sox9* expression in RPE. They provide a well-organized and logical set of experiments that provide a viable and interesting mechanism involving retinoic acid and *Sox9* expression in the RPE. The data looks solid, with good quality images. The mechanisms governing choroidal vascular development remain poorly understood, so the current study provides a novel and important advance to the field.

Although the study is overall very interesting and insightful, one slight weakness is that the manuscript only provides correlation of *Aldh1a1, Sox9*, and VEGF expression, and lacks the mechanistic evidence on how *Aldh1a1* regulates *Sox9* and therefore regulates VEGF expression in RPE cells.

Specific comments:

Although *Sox9* has been previously been reported to regulate VEGF expression, experiments showing the expression of VEGF in *Sox9* overexpressing mice and knockout mice would strengthen the conclusion of the manuscript.

The authors suggest that *Aldh1a1*-derived RA from dorsal neural retina upregulate *Sox9* and then VEGF in RPE. Can vitamin A diet rescue choroidal hyperplasia in *Aldh1a1* knockout mice?

The vascularized area (%) from both dorsal and ventral choroid should be plotted in Figure 5A and 5B.

*Reviewer #2:*

This paper reveals an interesting observation of choroid development in the mouse. By analyzing the *Aldh1a1^-/-^* mice the authors found a peculiar phenotype of choroidal hypoplasia only in the dorsal retina. Interestingly the ventral choroid is not dependent on RA signaling, as *Aldh1a3* conditional KO does not have an effect on choroid development. The authors suggest a model of the mechanism, where RA synthesized by *Aldh1a1* in the dorsal domain of the neural retina induces *Sox9* in the dorsal RPE that ultimately leads to increased secretion of VEGF from RPE, and subsequent regulation of choroid development. These findings contribute to our understanding of choroid development, suggesting that there are differential mechanisms for regulation of choroid development along the dorso-ventral (D/V) axis of the mouse retina.

1) The phenotype should be thoroughly investigated using comprehensive characterization of the choroid along thDV axis. EM analyses along the D/V axis in WT and *Aldh1a1^-/-^* at "critical" domains, such as dorsal, central and ventral, would improve the understanding of these processes.

2) The authors should do a better job explaining the phenotype in *Aldh1a1^-/-^*. How penetrant is the loss of pigmentation phenotype? Why is it patchy? And does it always appear in the same location or does it vary within the dorsal domain? In the retina shown in Figure 1D the unpigmented area seems to expand towards the central domain, outside of the *Aldh1a1* expression region. Is this correct? Are the other phenotypes, i.e., choroid thickness reduction and decrease in vascular area, also patchy or are they observed throughout the whole dorsal region, correlating with the domain of *Aldh1a1* expression? It would be helpful to correlate the appearance of phenotypes with the expression pattern of *Aldh1a1*. Along these lines, it is not clear what the authors think is the most affected layer of the choroid in the *Aldh1a1^-/-^*? The Haller's (large), Sattler's (medium) or choriocapillaris?

3) Due to the homeostatic regulation of RA signaling, the authors should check if the expression of *Aldh1a3* and *Cyp26s* were altered in the *Aldh1a1^-/-^* and vice-versa.

4) One concern relates to the foundations for the mechanistic conclusions of the paper. The model and main interpretations leading to the model are based on genetic manipulations using Cre-lines that the authors claim to be specific, i.e., *Tyr*-Cre, specific for RPE, and *Pax6*-α-Cre, specific for neural retina. However, since the specificities of these are critical for the conclusions of the paper, the authors need to better characterize them. Certainly, these lines have been published before but they were not described in the retina/choroid context. For instance, how specific is the *Tyr*-Cre line to the RPE? Since the choriocapillaris has melanin, it is possible that the manipulations of *Sox9*, either the KO or the overexpression rescue experiment, are not totally RPE-specific. This might affect the authors' interpretations in terms of the mechanism. Note that *Sox9* is also expressed in the choroid (Figure 4). Is *Pax6*-α-Cre expressed in all neural retina?

5) The phenotype in the *Sox9* RPE-conditional KO should be quantified, as in the *Aldh1a1^-/-^*, by measuring the% of vascular area. Likewise, RPE cell integrity should be assessed as well as the reduced levels of VEGF produced in the RPE-choroid complexes of the *Sox9* RPE-KO. Similarly, the phenotype in the rescue experiment of overexpression of RPE-*Sox9* in the *Aldh1a1^-/-^* should be quantified in the same way. It seems that it is only rescued to some extent. And are the secretion levels of VEGF also rescued?

- The authors claim that endomucin is a specific marker of choriocapillaris (Results subsection “*Aldh1a1*^-/-^ mice exhibited choroidal hypoplasia in the dorsal region”) but a previous paper shows that expression of endomucin extends to other layers of the choroid (Saint-Geniez et al., 2006). CD31 seems to be more specific to the choriocapillaris than endomucin (Saint-Geniez et al., 2006).

- At the end of the second paragraph in the Discussion, the authors mentioned that it is "possible that *Adh1a1* is also responsible for maintenance of the choriocapillaris". Their data on the VAD-diet experiment strongly suggests that indeed this is the case, as the VAD-diet was used starting from P0 and not applied in embryonic stages by feeding the pregnant mice.

- In the Figure 2 legend (C) mentions "orthogonal images of the Z-stacks (…) showing thin choroid…". It is actually hard to see the thinning of the choroid on these images. There is certainly more "breaks/holes" suggesting more avascular areas.

- Also, it is not clear if the RPE flat mount immunohistochemistry of the *Aldh1a1^-/-^* shown in Figure 2D is underlying a region of reduced choroid vascular area. Again, the authors need to describe better the phenotype patterns of *Aldh1a1^-/-^*.

- Could not find reference "Cohen Y. et al., 2016".

- In the Discussion, can the authors comment on the fact that there is a fairly large distinct domain in the mouse retina that lacks expression of *Aldh1a1* and *Aldh1a3* in the central domain, clearly visible in the RARE-lacZ transgenic mouse line. How is the choroid structure in this central domain? Is the choroid development still dependent on RA signaling at this region?

*Reviewer #3:*

This manuscript by Goto and coworkers evaluated the role of retinal ALDH1a1, an enzyme that synthesizes retinoic acid, in mediating dorsal choroidal vascular development in mice. The authors showed that mice with retinal neuronal deficiency of ALDH1a1 exhibited decreased VEGF and *Sox9* production in RPE (retinal pigment epithelium), and dorsal choroidal hypoplasia, which was phenocopied in mice with RA-deficient diet and also in mutant mice with RPE specific loss of *Sox9*. Based on these findings the authors concluded that neuronal ALDH1a1 affected dorsal choroidal vascular development via influencing retinoic acid-dependent, *Sox9*-mediated VEGF production in RPE. Overall this is a beautiful study that convincingly demonstrated the molecular interaction between retina and RPE via ALDH1a1/RA/*Sox9*/VEGF axis to control dorsal choroidal vascular development, which is of high interest for research in both basic development biology and also eye diseases with choroidal abnormalities such as age-related macular degeneration. This reviewer has just a few concerns as following.

1) There are patches of depigmentation in ALDH1a1 KO choroid/RPE (Figure 1D). Do ALDH1a1 KO eyes have any defects in melanin production in RPE melanosomes and choroidal melanocytes, in a potentially RA-dependent manner?

2) Reduced levels of *Sox9* in ALDH1a1 KO eyes were demonstrated with immunohistochemistry. It would be helpful to confirm this finding with Western blot and/or RT-PCR in isolated retinas/RPE. In addition, does RA treatment increase *Sox9* levels in RPE cell culture?

3) Figure 3G WT images also showed a small patch of choroidal thinning in top left part. Is this normal? Can the authors explain?

4) Figure 1C*Aldh1a1* staining does not completely overlap with GS staining, suggesting cells other than muller cells, e.g. photoreceptors, may also express *Aldh1a1*.

---

## [Author Response]

As you will see, all of the reviewers were impressed with the importance of your work, but they also had a number of specific comments.I am including the three reviews at the end of this letter, as there are a variety of specific and useful suggestions in them. Most of the comments are focused on greater documentation of the ocular phenotypes. One point that may be more challenging to address in a reasonable time frame is the specificity of the Tyr-Cre and Pax6-α-Cre lines. This can be done by crossing these Cre lines to a reporter (a nuclear localized reporter might be best). This point is summarized as item 4 in the comments of reviewer #2, which I am reproducing below."One concern relates to the foundations for the mechanistic conclusions of the paper. The model and main interpretations leading to the model are based on genetic manipulations using Cre-lines that the authors claim to be specific, i.e., Tyr-Cre, specific for RPE, and Pax6-α-Cre, specific for neural retina. However, since the specificities of these are critical for the conclusions of the paper, the authors need to better characterize them. Certainly, these lines have been published before but they were not described in the retina/choroid context. For instance, how specific is the Tyr-Cre line to the RPE? Since the choriocapillaris has melanin, it is possible that the manipulations of Sox9, either the KO or the overexpression rescue experiment, are not totally RPE-specific. This might affect the authors' interpretations in terms of the mechanism. Note that Sox9 is also expressed in the choroid (Figure 4). Is Pax6-α-Cre expressed in all neural retina?"

Thank you for your comments. We managed to obtain albinized *Tyr*-Cre and *Pax6*-α-Cre mice carrying the R26R-H2B-mCherry conditional reporter, which express nuclear mCherry in a Cre-dependent manner (Abe et al. 2011). We characterized mCherry expression at E16.5 because choroidal hypoplasia in *Aldh1a1^–/–^* mice was already observed at that time point, and we did not have enough time before the revision deadline to raise the pups.

In the *Tyr*-Cre; R26R-H2B-mCherry mice, mCherry expression was observed in all RPE cells. No mCherry expression was found in choroidal endothelial cells that were not co-immunostained with ERG and *Sox9* antibodies. We found some mCherry expression in the neural retinas, but few cells were double-labeled with *Sox9*. These results indicate that *Sox9* was conditionally disrupted in RPE cells in *Sox9*^RPE-KO^ mouse eyes. However, in *Pax6*-α-Cre and R26R-H2B-mCherry mice, mCherry expression was observed in the dorsal and ventral portions of retinal progenitor cells as described previously (Marquardt et al., 2001), and no mCherry expression was found in RPE cells.

Taken together, these results indicate that *Sox9* in RPE cells is responsible for dorsal choroidal vascularization. We added these data as Figure 6—figure supplement 1 and revised our manuscript as follows.

“To generate mice with selective deletion of *Sox9* in the developing RPE or neural retina, mice with a conditional deletion of *Sox9 (Sox9*^flox/flox^; Kist et al., 2002) were mated with either *Tyr*-Cre (*Sox9*^RPE-KO^) mice (Delmas et al., 2003) or *Pax6*-α-Cre (*Sox9*^Retina-KO^) mice (Marquardt et al., 2001), respectively[…] Also, mCherry expression in *Pax6*-α-Cre mice was restricted to the dorsal and ventral portions of the neural retina as reported previously (Marquardt et al., 2001), but no mCherry-positive cells were found in the choroid (Figure 6—figure supplement 1B).”

“Albinized R26R-H2B-mCherry mice (Abe et al., 2011) were used for the Cre reporter assay of *Tyr*-Cre and *Pax6*-α-Cre mice (Figure 6—figure supplement 1A and B).”

The challenge is made greater because the best design of this experiment would be to cross the Cre lines to reporters in an albino background (to avoid the difficulty of visualizing fluorescent reporters or immunofluorescent staining in heavily pigmented cells, i.e. RPE and choroid).Reviewer #1:[…] Although the study is overall very interesting and insightful, one slight weakness is that the manuscript only provides correlation of Aldh1a1, Sox9, and VEGF expression, and lacks the mechanistic evidence on how Aldh1a1 regulates Sox9 and therefore regulates VEGF expression in RPE cells.

We thank the reviewer for pointing this out. To address this question, we used cultured primary human RPE.

To demonstrate mechanistic evidence for the link between *Aldh1a1* and *Sox9*, we maintained the RPE cells with or without retinoic acids (RA), because RAs are synthesized by *Aldh1a1* and vitamin A deficient diet and administration of RA regulate choroidal vascular formation (Figure 4G–J). First, by ELISA, we confirmed a significant increase in the level of VEGF protein in the culture medium in response to RA exposure. We also quantified *Sox9* and *Vegfa* mRNAs, showing a significant RA-dependent increase. Subsequently, to examine whether *Sox9* regulates *Vegfa* expression in RPE cells, we performed overexpression and knockdown experiments. Overexpression of *Sox9* by transient transfection of pCAGIG-*SOX9* vector resulted in upregulation of *Vegfa* mRNA. In contrast, knockdown by *SOX9* siRNA transfection resulted in downregulation of *Vegfa* mRNA.

Based on these results, we concluded that *Sox9* enhanced by *Aldh1a1*-derived RA upregulates *Vegfa* expression in RPE cells. We added this result as Figure 5F–K and revised our manuscript as follows. We also replaced the data in Figure 4F with that obtained from the primary RPE experiments.

“To determine whether *Sox9* enhances VEGF in RPE cells in an RA-dependent manner, we measured *Sox9* and *Vegfa* mRNA expression in primary RPE cells in response to RA exposure. […] Taken together, these results strongly suggest that *Sox9* enhanced by Aldh1a1-mediated RA upregulates *Vegfa* expression in RPE cells.”

“Considering that in primary RPE cells RA exposure enhances both *Sox9* and *Vegfa* expression and that overexpression and knockdown of *Sox9* influences *Vegfa* expression, it is more likely that A*ldh1a1*-mediated RA production stimulates *Sox9* expression in dorsal RPE cells and *Sox9* then transactivates the *Vegfa* promoter.”

Specific comments:Although Sox9 has been previously been reported to regulate VEGF expression, experiments showing the expression of VEGF in Sox9 overexpressing mice and knockout mice would strengthen the conclusion of the manuscript.The authors suggest that Aldh1a1-derived RA from dorsal neural retina upregulate Sox9 and then VEGF in RPE. Can vitamin A diet rescue choroidal hyperplasia in Aldh1a1 knockout mice?

We thank the reviewer for these comments. We performed oral administration of RA to *Aldh1a1^-/-^* mice from E10 to E16 and found that the choroidal hypoplasia was rescued in P3 pups. We added the data in Figure 4I and J and revised our manuscript as follows.

“The level of VEGF in the RPE/choroid was significantly decreased in *Aldh1a1*^–/–^ mice, and RA-dependent enhancement of VEGF was observed in in vitro-cultured primary RPE cells. An RA-deficient diet resulted in dorsal choroidal hypoplasia, and simple RA treatment of *Aldh1a1*^–/–^ pregnant females suppressed choroid hypoplasia in their offspring.”

“Also, RA administration to *Aldh1a1^−/−^* mice by oral gavage of pregnant mothers from E10 to E16 significantly suppressed the dorsal choroidal hypoplasia (Figure 4I and J). These results indicate that RA controls dorsal choroidal vascular development and that dorsal choroidal hypoplasia in *Aldh1a1^−/−^* mice is causally related to a RA synthesis deficiency.”

The vascularized area (%) from both dorsal and ventral choroid should be plotted in Figure 5A and 5B.

We thank the reviewer for these comments. We added the appropriate graphs to Figure 6B and D and revised our manuscript as follows.

“In *Sox9*^RPE-KO^ mice, we found less pigmentation in the dorsal region, and significantly poorer vasculature in the dorsal choroidal area than the other areas (Figure 6A and B),”.

*“Tyr*-Cre-induced overexpression of *Sox9* in these mice significantly recovered the choroidal hypoplasia phenotypes in the dorsal region (Figure 6C and D).”

Reviewer #2:[…] 1) The phenotype should be thoroughly investigated using comprehensive characterization of the choroid along thDV axis. EM analyses along the D/V axis in WT and Aldh1a1^-/-^ at "critical" domains, such as dorsal, central and ventral, would improve the understanding of these processes.

Thank you for your suggestion. We performed EM analysis in 8-week-old WT and *Aldh1a1^–/–^* mouse eyes. As shown by the H&E staining (Figure 1E), EM sections of the dorsal *Aldh1a1^–/–^* eyes also showed a thinner choroidal layer and less pigmentation than those of the other experimental samples (dorsal and ventral WT eyes and ventral *Aldh1a1^–/–^* eyes). Despite the observation that the choroidal blood vessels in *Aldh1a1^–/–^* mice exhibited morphological and immunohistochemical abnormalities, they have fenestrations, which is a characteristic structure of choriocapillaris. We added these data as Figure 3A–C and revised our manuscript as follows.

“To investigate further the morphological features of the choroidal vasculature and the neural retina/RPE/Bruch’s membrane complex, sections of WT and *Aldh1a1*^–/–^ eyes were examined by transmission electron microscopy (TEM). […] Taken together, these results suggest that the hypoplastic blood vessels of the *Aldh1a1*^–/–^ eyes maintain the characteristics of the choriocapillaris, including an intact RPE and Bruch’s membrane.”

The central domain of choroidal flat-mount showed the same density of vascular area in both dorsal and ventral areas of WT and the ventral area of *Aldh1a1^–/–^* choroid. We refer the reviewer to Author response image 1 below, which shows the central part of the choroid in WT and *Aldh1a1^–/–^*mice.

**Author response image 1. respfig1:** Representative choroidal flat-mount IHC of the central region of adult (8-week-old) WT (left panel) and *Aldh1a1*^–/–^ (right panel) eyes immunostained with FITC-labeled isolectin B4 (IB4, green) and anti-endomucin antibody (Emcn, red). There was no difference in choroidal vascular density between WT and Aldh1a1^–/–^ mice.

2) The authors should do a better job explaining the phenotype in Aldh1a1^-/-^. How penetrant is the loss of pigmentation phenotype? Why is it patchy? And does it always appear in the same location or does it vary within the dorsal domain? In the retina shown in Figure 1D the unpigmented area seems to expand towards the central domain, outside of the Aldh1a1 expression region. Is this correct?

We thank the reviewer for these comments. We first became interested in the phenotype. However, we could not identify an appropriate genotype–phenotype relationship. In the mating record, as shown in Author response image 2, not all *Aldh1a1^–/–^* mice had a loss-of-pigmentation phenotype. Also, some WT mice had the same phenotype as observed in *Aldh1a1^–/–^* mice. This observation suggests that multiple factors besides *Aldh1a1* cause the loss-of-pigmentation phenotype. However, all *Aldh1a1^–/–^* mice have dorsal choroidal hypoplasia. For this reason, we just described the loss-of-pigmentation phenotype and did not proceed with further analyses.

**Author response image 2. respfig2:** The frequency of appearance of pigmentation loss did not correspond with the principles of Mendelian inheritance.

Are the other phenotypes, i.e., choroid thickness reduction and decrease in vascular area, also patchy or are they observed throughout the whole dorsal region, correlating with the domain of Aldh1a1 expression? It would be helpful to correlate the appearance of phenotypes with the expression pattern of Aldh1a1.

As the reviewer pointed out, choroidal hypoplasia in *Aldh1a1^–/–^* mice is observed in a dorsal portion of the choroid, where *Aldh1a1* is to be expressed in the WT neural retina (see Figure 2A and Figure 1—figure supplement 1B). This observation indicates that neural retina-specific expression of *Aldh1a1* controls dorsal choroidal vascular development.

Along these lines, it is not clear what the authors think is the most affected layer of the choroid in the Aldh1a1^-/-^? The Haller's (large), Sattler's (medium) or choriocapillaris?

Our understanding is that all types of choroidal blood vessels in the dorsal region are affected in *Aldh1a1^–/–^* mice. In fact, Figure 2B indicates that the hypoplasia phenotype is observed not only in choriocapillaris (endomucin-positive), but also in Haller/Sattler vessels (IB4-positive). Also, as shown in Figure 3—figure supplement 1, differential expression of VEGFR1 and VEGFR2 in choroidal vessels disappeared, suggesting that all types of dorsal choroidal vessels are immature in *Aldh1a1^–/–^* mice.

It might be considered that choriocapillaris is most affected because the more vascular region of choriocapillaris decreased more than the Haller/Sattler vessels. However, TEM images indicated that the choriocapillaris in *Aldh1a1^–/–^* dorsal choroid still had fenestrations, suggesting that the choriocapillaris retains its physiological characteristics.

3) Due to the homeostatic regulation of RA signaling, the authors should check if the expression of Aldh1a3 and Cyp26s were altered in the Aldh1a1^-/-^ and vice-versa.

We thank the reviewer for these comments. Spatiotemporal analysis of RA production in *Aldh1a1^–/–^* mouse eyes has been performed previously using RARE-LacZ mice, which showed that the RA-free zone was expanded to include the dorsal area (Fan et al., 2003; Matt et al., 2005). These observations indicate that RA is not synthesized in the dorsal region in *Aldh1a1^–/–^* mice. Also, we performed immunohistochemistry (IHC) to check the compensatory effect of *Aldh1a3* in E17.5 *Aldh1a1^–/–^* eyes. The results showed that the spatial distribution of expression of *Aldh1a3* was not altered (Author response image 3), indicating no compensatory effect of *Aldh1a3*.

**Author response image 3. respfig3:** Section IHCs of E17.5 mouse eyes labeled with anti-Aldh1a3 (green) and anti-*Aldh1a1* (red) antibodies. The region of *Aldh1a3* expression did not expand.

4) One concern relates to the foundations for the mechanistic conclusions of the paper. The model and main interpretations leading to the model are based on genetic manipulations using Cre-lines that the authors claim to be specific, i.e., Tyr-Cre, specific for RPE, and Pax6-α-Cre, specific for neural retina. However, since the specificities of these are critical for the conclusions of the paper, the authors need to better characterize them. Certainly, these lines have been published before but they were not described in the retina/choroid context. For instance, how specific is the Tyr-Cre line to the RPE? Since the choriocapillaris has melanin, it is possible that the manipulations of Sox9, either the KO or the overexpression rescue experiment, are not totally RPE-specific. This might affect the authors' interpretations in terms of the mechanism. Note that Sox9 is also expressed in the choroid (Figure 4). Is Pax6-α-Cre expressed in all neural retina?

We thank the reviewer for these comments. Please see the answer to the editor comments.

5) The phenotype in the Sox9 RPE-conditional KO should be quantified, as in the Aldh1a1^-/-^, by measuring the% of vascular area. Likewise, RPE cell integrity should be assessed as well as the reduced levels of VEGF produced in the RPE-choroid complexes of the Sox9 RPE-KO. Similarly, the phenotype in the rescue experiment of overexpression of RPE-Sox9 in the Aldh1a1^-/-^ should be quantified in the same way. It seems that it is only rescued to some extent. And are the secretion levels of VEGF also rescued?

As you suggested, we added quantification data to Figure 6B and D and revised our manuscript (please see the last part of the answer for specific comments from reviewer #1). Since we were not able to obtain sufficient *Sox9*^RPE-KO^ and *Aldh1a1^–/–^;Sox9*^RPE-OE^ mice to compare the VEGF secretion levels, we instead quantified VEGF expression in cultured primary RPE cells in response to overexpression and knockdown of *SOX9*. Please see the answer to reviewer #1’s main comments.

- The authors claim that endomucin is a specific marker of choriocapillaris (Results subsection “Aldh1a1^-/-^ mice exhibited choroidal hypoplasia in the dorsal region”) but a previous paper shows that expression of endomucin extends to other layers of the choroid (Saint-Geniez et al., 2006). CD31 seems to be more specific to the choriocapillaris than endomucin (Saint-Geniez et al., 2006).

Besides the rat anti-endomucin antibody and FITC-labeled isolectin B4 used in this study, we had checked the specificity of rat anti-CD-31 antibody before starting this study. Because these markers had been tested previously only in section immunohistochemistry, we performed flat-mount immunohistochemistry to figure out which marker explicitly discriminated choriocapillaris from choroidal medium-sized/large vessels. As shown in Author response image 4, in adult WT mice (8 weeks old), anti-CD-31 antibody mostly visualized medium-sized/large vessels of the choroid that were double-labeled with Isolectin B4, but hardly visualized choriocapillaris. In contrast, anti-endomucin antibody specifically labeled choriocapillaris that was not double-labeled with Isolectin B4. Based on these preliminary experiments, we chose the anti-endomucin antibody to visualize and quantify choriocapillaris vascularization.

**Author response image 4. respfig4:** The representative dorsal region of choroidal flat-mounts of 8-week-old WT and *Aldh1a1^–/–^* mice stained with FITC-isolectin B4 (IB4, green), anti-endomucin (Emcn, red; upper panels), and anti-CD-31 (red; lower panels). Emcn specifically visualized choriocapillaris, whereas CD-31 visualized medium-sized/large vessels that were IB4-positive.

- At the end of the second paragraph in the Discussion, the authors mentioned that it is "possible that Adh1a1 is also responsible for maintenance of the choriocapillaris". Their data on the VAD-diet experiment strongly suggests that indeed this is the case, as the VAD-diet was used starting from P0 and not applied in embryonic stages by feeding the pregnant mice.

We assessed VAD pups using a procedure described previously (Chihara et al., 2013) and quantified the vascularization from choroidal flat-mounts immunostained with anti-endomucin antibody. The results showed that P3 VAD pups phenocopied the dorsal choroidal hypoplasia observed in *Aldh1a1^–/–^* mice, indicating that RA is responsible for dorsal choroidal vascularization. We added the results to Figure 4G and H and revised our manuscript as follows.

“At P3, VAD mice showed dorsal choroidal hypoplasia in the flat-mount analysis (Figure 4G).”

- In the Figure 2 legend (C) mentions "orthogonal images of the Z-stacks (…) showing thin choroid…". It is actually hard to see the thinning of the choroid on these images. There is certainly more "breaks/holes" suggesting more avascular areas.

Thank you for your suggestion, we revised the Figure 2C legend as follows.

“Orthogonal images of the Z-stacks (broken lines in (B)) showing breaks/holes in the dorsal region of *Aldh1a1*^–/–^ eyes stained with Emcn antibody (red) and IB4 (green).”

- Also, it is not clear if the RPE flat mount immunohistochemistry of the Aldh1a1^-/-^ shown in Figure 2D is underlying a region of reduced choroid vascular area. Again, the authors need to describe better the phenotype patterns of Aldh1a1^-/-^.

Author response image 5 shows X-Y images of the dorsal region of WT and *Aldh1a1^–/–^* choroidal flat mounts immunostained with anti-endomucin and anti-ZO-1 antibodies. Thinner choriocapillaris formation was observed in *Aldh1a1^–/–^* mouse eyes, but RPE morphology visualized by ZO-1 appeared normal.

**Author response image 5. respfig5:** Representative dorsal choroidal flat-mounts of adult (8-week-old) WT (left panel) and *Aldh1a1*^–/–^ (right panel) eyes immunostained with anti-endomucin (Emcn, red) and anti-ZO-1 (green) antibodies. There is no difference in the size of RPE between WT and *Aldh1a1^–/–^* mice.

- Could not find reference "Cohen Y. et al., 2016".

Thank you for the comment. We mentioned URL of ARVO abstract as follows.

“Cohen Y, Blinder P, Idelson M, Reubinoff B, Itzkovitz S, Ashery-Padan R. 2016. Pax6 role in the regulation of retinal pigmented epithelium maturation. Invest Ophthalmol Vis Sci 57: 6055. http://iovs.arvojournals.org/article.aspx?articleid=2563982.”

- In the Discussion, can the authors comment on the fact that there is a fairly large distinct domain in the mouse retina that lacks expression of Aldh1a1 and Aldh1a3 in the central domain, clearly visible in the RARE-lacZ transgenic mouse line. How is the choroid structure in this central domain? Is the choroid development still dependent on RA signaling at this region?

We thank the reviewer for these comments. Please see the response to the major comment by reviewer #2.

Reviewer #3:[…] This reviewer has just a few concerns as following.1) There are patches of depigmentation in ALDH1a1 KO choroid/RPE (Figure 1D). Do ALDH1a1 KO eyes have any defects in melanin production in RPE melanosomes and choroidal melanocytes, in a potentially RA-dependent manner?

Thank you for the comment. Please read the answer to reviewer #2’s major comment 2.

2) Reduced levels of Sox9 in ALDH1a1 KO eyes were demonstrated with immunohistochemistry. It would be helpful to confirm this finding with Western blot and/or RT-PCR in isolated retinas/RPE. In addition, does RA treatment increase Sox9 levels in RPE cell culture?

Please see the answer to reviewer #1’s main comment and reviewer #2’s major comment 5.

3) Figure 3G WT images also showed a small patch of choroidal thinning in top left part. Is this normal? Can the authors explain?

A small patch of the choroid is normal, which is why the choroidal vascular densities of WT are around 90% (Figures 2A and 6B).

4) Figure 1C Aldh1a1 staining does not completely overlap with GS staining, suggesting cells other than muller cells, e.g. photoreceptors, may also express Aldh1a1.

We repeated and revised the IHC shown in Figure 1C. We also added high magnification images of section ISH for *Aldh1a1* mRNA as Figure 1—figure supplement 1A, showing that the ISH signals were restricted to the middle of the INL, where some Muller glia were localized. No ISH signals were found in the ONL.